



# Improvement of numerical weather prediction model analysis during fog conditions through the assimilation of ground-based microwave radiometer observations: a 1D-Var study.

Pauline Martinet[a], Domenico Cimini[b], Frédéric Burnet[a], Benjamin Ménétrier[a], Yann Michel[a], and Vinciane Unger[a]

[a]CNRM UMR 3589, Meteo-France/CNRS, Toulouse, France
[b]IMAA-CNR, Potenza, Italy

**Correspondence:** Pauline Martinet (pauline.martinet@meteo.fr)

**Abstract.** This paper investigates the potential benefit of ground-based microwave radiometers (MWRs) to improve the initial state (analysis) of current numerical weather prediction (NWP) systems during fog conditions. To that end, temperature, humidity and liquid water path (LWP) retrievals have been performed using a one-dimensional variational technique (1D-Var) during a fog dedicated field-experiment performed over winter 2016-2017 in France. In-situ measurements from a 120 m tower and radiosoundings are used to assess the improvement brought by the 1D-Var analysis to the background. A sensitivity study demonstrates the importance of the cross-correlations between temperature and specific humidity in the background-error-covariance matrix as well as the bias-correction applied on MWR raw measurements. With the optimal 1D-Var configuration, a root-mean-square error smaller than 1.5 K (resp. 0.8 K) for temperature and 1 g.kg$^{-1}$ (resp. 0.5 g.kg$^{-1}$) for humidity is obtained up to 6 km altitude (resp. within the fog layer up to 250 m). A thin-radiative fog case study has shown that the assimilation of MWR observations was able to correct large temperature errors of the AROME model as well as vertical and temporal errors observed in the fog lifecycle. During missed fog profiles, 1D-Var increments pull towards lower temperature close to the ground and higher temperature above 100 m altitude, i.e. higher atmospheric stability. The largest analysis increments and background errors are observed during false alarms when the AROME forecasts tend to significantly overestimate the temperature cooling. The impact on specific humidity was found neutral to slightly positive. The impact on LWP was found significant with 1D-Var increments within 200 g.m$^{-2}$ and RMSE with respect to MWR statistical regressions decreased from 101 g.m$^{-2}$ in the background to 27 g.m$^{-2}$ in the 1D-Var analysis. These encouraging results led to the deployment of 8 MWRs during the international SOFOG3D (SOuth FOGs 3D experiment for fog processes study) experiment conducted by Météo-France.



# 1 Introduction

Each year large human and economical losses are due to fog episodes, which, by the large reduction of visibility, affect aviation, marine, and land transportation (Gultepe et al. (2007)). Fog forecasts remain quite inaccurate due to the complexity, non linearities and fine scale of the physical processes taking part in the fog lifecycle. Fog results from a combination of radiative, turbulent and microphysical processes as well as interactions with surface heterogeneities which will drive the relative importance of local and large-scale circulations. Recently, three dimensional models have replaced one-dimensional models to forecast fog in most national weather services. Currently, convective-scale numerical weather prediction (NWP) models run with an horizontal resolution of approximately one kilometre with frequent data assimilation cycles. While the importance of vertical resolution (Philip et al. (2016)), aerosol activation (Mazoyer et al. (2019)) or water deposition (Tav et al. (2018)) have recently been highlighted to improve fog forecasts, fog is also known to be highly sensitive to initial conditions (Bergot and Guedalia (1994), Bergot et al. (2005), Hu et al. (2014)). Therefore, accurate initial temperature, humidity and wind profiles are crucial to successfully forecast fog. However, the atmospheric boundary layer (ABL) has also been identified as a part of the atmosphere which is undersampled in observations. Even if satellite data provide a global coverage all over the world, they provide limited information on the ABL due to the attenuation by clouds, the degraded vertical resolution in the ABL and the complexity of data assimilation over lands due to uncertainties in surface properties (skin temperatures, albedo, Guedj et al. (2011)). Recently, an Observing System Simulation Experiment (OSSE) by Hu et al. (2017), has demonstrated that temperature and moisture at the surface have a larger impact on fog forecast than surface wind observations concluding that temperature and humidity profilers could potentially play a major role in the improvement of fog forecast initialization. Ground-based microwave radiometers (MWR) are robust instruments providing continuous observations of temperature and humidity profiles as well as integrated liquid and water contents during all-sky weather conditions. Even if their vertical resolution degrades with altitude (Cimini et al. (2006)), most of their information content resides in the ABL (Löhnert and Maier (2012)) and their high temporal resolution (few minutes) makes them suitable to monitor the evolution of fog development. Despite the potential impact of MWRs in NWP models, assimilation experiments of their data have been limited to few attempts. The first preliminary study of Vandenberghe and Ware (2002) has demonstrated a positive impact of the assimilation of a single MWR unit into the 10 km horizontal resolution MM5 (https://www2.mmm.ucar.edu/mm5/) mesoscale model in the context of a winter fog event. The impact of a simulated network of 140 MWRs was also investigated by Otkin et al. (2011) and Hartung et al. (2011) on a winter storm case. This study confirmed a positive impact on temperature and humidity analyses as well as up to 12 hour forecasts on moisture flux. More recently, a real network of 13 MWRs was assimilated by Caumont et al. (2016) into the 2.5 km horizontal resolution convective scale model AROME in the context of heavy-precipitation events in the western Mediterranean. The impact of this network was found neutral on temperature and humidity fields but positive on quantitative precipitation forecasts up to 18 hours. In addition, Martinet et al. (2015) and Martinet et al. (2017) have demonstrated the positive impact that could be expected on NWP temperature profile analyses by the direct assimilation of MWR brightness temperatures into the AROME model thanks to a one-dimensional variational framework (1D-Var). All these studies showed an encouraging positive impact of the assimilation of MWRs observations into NWP, though they are limited to deep-convection, single case studies on low



resolution limited area models, or restricted to temperature analyses only. The purpose of this article is to evaluate the expected benefit of MWRs on km-scale NWP analyses during fog events on an extended dataset over a six-month fog experiment. This

expands the studies of Martinet et al. (2015) and Martinet et al. (2017) to humidity and liquid water path retrievals and evaluating the impact of new tools developed to optimize the assimilation of MWRs during COST actions TOPROF (Illingworth et al. (2019)) and PROBE (Cimini et al. (2020)). A fog dedicated field experiment was carried out in the North-East of France during the winter 2016-2017 during which a 14-channel MWR has been operated. The impact of MWR brightness temperatures on temperature, humidity and liquid water content profiles forecast by AROME has been evaluated during the six-month period

against in-situ data collected during intensive observation periods (IOPs) and continuous measurements deployed on a 120 m instrumented tower. This paper begins with an overview of the dataset, the AROME model and a description of the 1D-Var settings in section 2. A sensitivity study of the 1D-Var retrievals to the background-error-covariance matrix and bias-correction to select the optimal configuration is presented in section 3. Section 4 presents a case study of the first IOP showing large AROME errors during a thin radiative fog event well corrected by the 1D-Var. Section 5 generalizes the results obtained in

section 4 through a statistical evaluation of 1D-Var analysis errors and increments. Section 6 presents the deployment of a regional-scale MWR network for fog forecast improvement as continuity of this study, while finally section 7 summarises the main conclusions.

## 2   Dataset and methodology

### 2.1   Instrumentation

Data sampled during a field experiment dedicated to fog process studies carried out at the ANDRA (the French National Radioactive Waste Management Agency) atmospheric platform located in Houdelaincourt (48.5623N; 5.5055E) in the North-East of France during the winter 2016-2017 are used in this study. The experimental site was chosen due to the high occurence of fogs and the possibility to take advantage of a 120 m instrumented tower. A large range of in-situ instrumentation was deployed during the six-month experiment: visibility sensors, liquid water content and droplet size distribution measurements,

temperature and relative humidity measurements at different levels above ground (10 m, 50 m, 120 m). In addition to in-situ measurements, a 14-channel HATPRO MWR (Rose et al. (2005)) manufactured by Radiometer Physics Gmbh (RPG) was deployed on site during the experiment. The HATPRO MWR is a passive instrument measuring the naturally emitted downwelling radiance in two spectral ranges: 22.24 to 31 GHz to retrieve humidity profiles with a low resolution but high accurate integrated water contents (IWV) and liquid water path (LWP). The 51 to 58 GHz range, located in the 60 GHz $O_2$

absorption complex line, is used to retrieve temperature profiles. Elevation scans from 5.4° to 90° were used to improve the vertical resolution of temperature profiles assuming that horizontal homogeneity in the vicinity of the instrument is respected. A ceilometer Vaisala CL31 was deployed during October to December 2016 replaced by a Vaisala CT25K from January to April 2017 to determine the cloud base altitude. In addition, 21 VAISALA RS92 radiosondes with an expected accuracy of 0.5 K in temperature and 5 % in relative humidity were launched during IOPs. Tethered balloon measurements were also carried

out with the deployment of a cloud particle probe and a turbulence probe.



## 2.2 The AROME NWP model

In this study 1-hour forecasts from the French convective scale model AROME (Application of Research to Operations at MEsoscale, Seity et al. (2011)) are used as *a priori* profiles or "backgrounds". AROME is a limited area model covering Western Europe with non-hydrostatic dynamical core. Since beginning 2015, the horizontal resolution of AROME has been increased from 2.5 km to 1.3 km as well as the number of vertical levels from 60 to 90 (Brousseau et al. (2016)). Vertical levels follow the terrain in the lowest layers and isobars in the upper atmosphere. The detailed physics of Arome are inherited from the research Meso-NH model (Lafore et al. (1997)). Deep convection is assumed to be resolved explicitly, but shallow convection is parameterized following Pergaud et al. (2009). A bulk one-moment microphysical scheme (Pinty and Jabouille (1998)) governs the equations of the specific contents of six water species (humidity, cloud liquid water, precipitating liquid water, pristine ice, snow, and graupel). This new version also performs 3D-Var analyses every hour instead of every three hours to optimize the use of frequent observations. All conventional observations are assimilated together with wind profilers, winds from space-borne measurements (Atmospheric Motion Vectors and scatterometers), Doppler winds (Montmerle and Faccani (2009)) and reflectivity (Wattrelot et al. (2014)) from ground-based weather radars, satellite radiances as well as ground-based GPS measurements (Mahfouf et al. (2015)).

## 2.3 1D-Var framework

To retrieve temperature and humidity profiles and evaluate the impact on AROME analyses, a 1D-Var framework similar to the one described in Martinet et al. (2017) is used. Based on the optimal estimation theory by Rodgers (2000), MWR observations are optimally combined with an *a priori* estimation of the atmospheric state which, in this study, refers to 1-hour AROME forecasts. To that end, the two sources of information are weighted by corresponding uncertainty called the background-error-covariance matrix (**B**) for the *a priori* profile and the observation-error-covariance matrix (**R**) for the observation to find the optimal state. In order to find the optimal state minimizing the distance to the observation, a radiative transfer model is needed to compute the equivalent observation from the *a priori*. The method iteratively modifies the state vector $x$ from the *a priori* $x_b$ to minimize the following cost function:

$$J(\mathbf{x}) = \frac{1}{2}(\mathbf{x} - \mathbf{x}_b)^{\mathrm{T}}\mathbf{B}^{-1}(\mathbf{x} - \mathbf{x}_b) + \frac{1}{2}(\mathbf{y} - \mathrm{H}(\mathbf{x}))^{\mathrm{T}}\mathbf{R}^{-1}(\mathbf{y} - \mathrm{H}(\mathbf{x}))$$

where H represents the observation operator (radiative transfer model and interpolations from model space to observation space), $^{\mathrm{T}}$ represents the transpose operator and $^{-1}$ the inverse operator. The observation-error-covariance matrix **R** should take into account representativeness and forward model errors as well as radiometric noise.

For the first time, the fast radiative transfer model RTTOV-gb (De Angelis et al. (2016), Cimini et al. (2019)), developed specifically to simulate MWR observations for operational applications during the Cost action TOPROF, is used within the 1D-Var package maintained by the NWP Satellite Application Facility (https://www.nwpsaf.eu/site/software/1d-var/). To that end, the





1D-Var has been adapted to the ground-based sensing configuration of MWRs and interfaced with RTTOV-gb. In this study the control vector $x$ consists in temperature and the natural logarithm of specific humidity on the same 90 levels as defined in AROME. These levels cover the atmospheric range from the ground up to 30 km, the vertical resolution decreasing with

altitude: 20-100 m below 1 km, 100-200 m from 1 to 5 km, around 400 m at 10 km. Additionally to temperature and humidity, the liquid water path is also included in the control vector. Following the current implementation of the NWPSAF 1D-Var, no correlation between the LWP and the other variables is assumed in the **B** matrix. The observation vector $y$ consists in brightness temperatures (BT) in all K-band and V-band channels ([1]) at zenith and only opaque channels (above 54 GHz) at low elevation angles: 42°, 30°, 19.2°, 10.2° and 5.4 °.

## 3  Evaluation of 1D-Var retrievals

### 3.1  Background errors

In variational data assimilation (either 1D-Var or 3D/4D-Var), the accuracy of the analysis will depend on the background-error covariance matrix **B**. This matrix specifies how much weight is given to the *a priori* profile compared to the observation, how the information from the localized observation is spread in the model space both vertically and spatially (for 3D/4D-

Var assimilation) and impose the balance between the model control variables. However, due to difficulties in measuring the "true" state, this **B** matrix has to be modelled. Currently, climatological, spatially homogeneous and isotropic background-error covariances are used operationally in the AROME model (Brousseau et al. (2011)). They are computed from 3h range forecast differences from an ensemble data assimilation over long time periods and the whole model domain, making the couplings between variables as well as vertical correlations inadequate for fog areas. For this study, a similar approach as the

one described in Ménétrier and Montmerle (2011) has been used to infer background-error covariances adapted to fog layers and to the AROME configuration and the time period of the experiment. To that end, the AROME ensemble data assimilation schemes (AROME EDA) that mimics in a variational context the approach taken in the stochastic Ensemble Kalman Filter (Evensen (2003)) has been used. The EDA explicitly perturbs the observations, the model and the boundary conditions, and gives in return estimates of analysis and background error covariance (Fisher (2003); Zagar et al. (2005)). The AROME EDA

consists in running an ensemble of 3D-Vars in parallel, where the observations are perturbed according to their prescribed error statistics. The model perturbations are represented by an online multiplicative inflation scheme (Raynaud and Bouttier, 2015). The inflation factor is derived from the skill over spread ratio. The perturbed boundary conditions are taken from the global EDA (Raynaud et al., 2011). The EDA configuration used for this study corresponds to the operational implementation since July 2018 with a spatial resolution finally set to 3.2 km and an ensemble size of 25 members.

Firstly, using this AROME EDA, a so-called "climatological" **B** was obtained by computing the forecast differences $\epsilon_b^{k,l} = x_b^k - x_b^l$ between members k,l for all grid points of the whole AROME domain and all assimilation cycles on the 28th of October 2016 (IOP1). A specific fog **B** matrix was then computed by applying a fog mask in order to only select grid points for which most

---

[1] 22.24, 23.04, 23.84, 25.44, 26.24, 27.84, 31.4, 51.26, 52.28, 53.86, 54.94, 56.66, 57.3 and 58 GHz





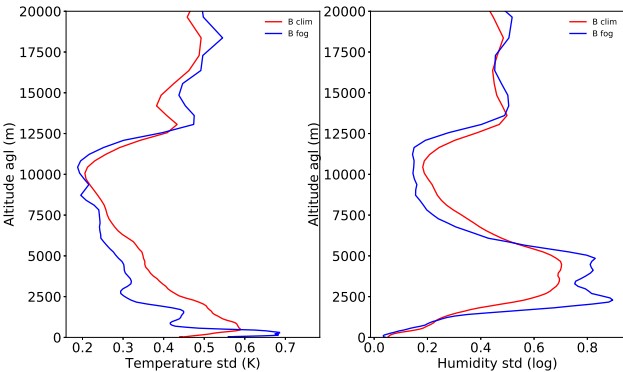

**Figure 1.** Background error standard deviations for temperature (left pannel) and the natural logarithm of specific humidity (right panel) for a "climatological" **B** matrix (red line) or a specific fog **B** matrix (blue line).

of the EDA members forecast fog. According to the discussion on the fog-model predictor used in Ménétrier and Montmerle (2011), the fog mask was based on the presence of liquid water contents above $10^{-6}$ kg.kg$^{-1}$ in the first three layers of the

model. Several tests have been performed using 3h range forecasts from different assimilation cycles to finally retain the fog **B** matrix showing the best results in terms of root-mean-square-errors (RMSE) with respect to radiosoundings. Similarly to Brousseau et al. (2016), background error standard deviations are multiplied by a factor $\alpha < 1$ in order to take into account the forecast error reduction while the background range decreases from 3 to 1h (as the AROME EDA provides 3h forecasts whereas the 1D-Var deals with 1h forecasts). Based on comparison with in-situ measurements, an optimal value of $\alpha = 0.7$ was

found. This multiplicative factor is only applied on background error standard deviations while cross-correlations are assumed to be the same at the 1 and 3h forecast ranges. Figure 1 compares background error standard deviations for temperature and the natural logarithm of specific humidity computed for the "climatological" and "fog" **B** matrices. Similar shape and magnitude are observed between the two **B** matrices for the natural logarithm of specific humidity. However, in the case of temperature, background errors in fog areas are found to be larger within the first 500 m with a maximum of 0.7 K at 250 m. On the other

hand, the "climatological" **B** matrix shows values below 0.5 K within the whole fog layer. Figure 2 shows the cross-correlations between specific humidity and temperature. Similarly to Ménétrier and Montmerle (2011), a strong positive coupling appears in the fog layer within the first 200 m. This coupling implies that a positive temperature error will be translated into a positive specific humidity error (and vice-versa) due to saturated conditions. This structure significantly differs from the one observed in climatological conditions with almost no coupling between the two variables in the boundary layer. The fog layer is also

un-coupled with atmospheric layers above the fog top which exhibit a negative coupling between temperature and humidity.

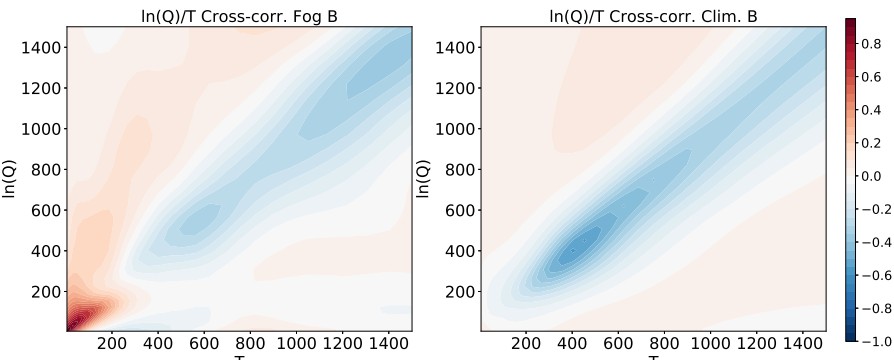

**Figure 2.** Cross-correlations between the natural logarithm of specific humidity (y-axis) and temperature (x-axis) for a fog **B** matrix (left panel) or a climatological **B** matrix (right panel). x-axis and y-axis are labelled according to altitude above ground in meter.

## 3.2 Optimal configuration of 1D-Var retrievals

The accuracy of 1D-Var retrievals depends, not only on the background-error-covariance matrix but also on the good specification of the observation-error-covariance matrix. Observation errors are assumed to follow Gaussian distributions with zero mean. A similar method as described in Martinet et al. (2015), De Angelis et al. (2017) and Cimini et al. (2020) has been used to implement a bias correction of BT measurements based on 6-month differences between MWR observations and BTs simulated from AROME 1h forecasts with the use of RTTOV-gb (so-called "O-B monitoring"). Table 1 reports the biases obtained for each channel at 90° and the most opaque channels at low elevation angles (transparent channels are not used at low elevation angles due to the violation of the assumption of horizontal homogeneity). The values are consistent with those reported in De Angelis et al. (2017). A static bias-correction of all channels based on table 1 has been applied to the measurements. Observation errors due to liquid nitrogen calibration and spectroscopic errors in radiative transfer models were updated according to recent studies from Maschwitz et al. (2013) and Cimini et al. (2018). Therefore, in addition to commonly used values of instrumental noise (0.5 K for transparents channels and 0.2 K for the most opaque channels), the individual errors defined by Maschwitz et al. (2013) and Cimini et al. (2018) were added in quadrature. It is important to note that calibration errors of modern MWRs are lower than the ones used in this study due to new developments in the manufacturer software and liquid nitrogen target used for the radiometer calibration. Table 2 summarizes the total observation uncertainty for each channel.

In order to define the best configuration of 1D-Var retrievals in terms of background-error-covariance matrix and bias-correction, statistics have been performed over the 6-month period by comparison with the 120-m tower measurements. For each altitude instrumented with a weather station (50 and 120 m altitude) and each variable (temperature and specific humidity), the error reduction brought by the analysis over the background is defined as:

$$ER = 1 - \frac{RMSE_{xa}}{RMSE_{xb}}$$





**Table 1.** Bias of the observation minus background departures computed from AROME forecasts for all frequency at 90° elevation angle and only the most opaque channels (54.94 to 58 GHz) at lower elevation angles.

|  | 22.24 | 23.04 | 23.84 | 25.44 | 26.24 | 27.84 | 31.4 | 51.26 | 52.28 | 53.86 | 54.94 | 56.66 | 57.3 | 58 |
|---|---|---|---|---|---|---|---|---|---|---|---|---|---|---|
| 90 ° | 0.41 | 0.66 | 0.17 | 0.18 | 0.15 | -0.43 | 0.31 | -1.30 | -4.72 | -0.28 | -0.04 | 0.06 | 0.16 | 0.20 |
| 42 ° | - | - | - | - | - | - | - | - | - | - | 0.04 | 0.18 | 0.22 | 0.23 |
| 30 ° | - | - | - | - | - | - | - | - | - | - | 0.07 | 0.24 | 0.27 | 0.27 |
| 19.2 ° | - | - | - | - | - | - | - | - | - | - | 0.14 | 0.31 | 0.33 | 0.30 |
| 10.2 ° | - | - | - | - | - | - | - | - | - | - | 0.23 | 0.37 | 0.35 | 0.32 |
| 5.4 ° | - | - | - | - | - | - | - | - | - | - | 0.18 | 0.24 | 0.25 | 0.21 |

**Table 2.** Observation uncertainties (K) prescribed in the observation-error-covariance matrix for each channel.

| Frequency (GHz): | 22.24 | 23.04 | 23.84 | 25.44 | 26.24 | 27.84 | 31.4 | 51.26 | 52.28 | 53.86 | 54.94 | 56.66 | 57.3 | 58 |
|---|---|---|---|---|---|---|---|---|---|---|---|---|---|---|
| $\sigma_o$ (K): | 1.34 | 1.71 | 1.16 | 1.08 | 1.25 | 1.17 | 1.19 | 3.21 | 3.29 | 1.30 | 0.37 | 0.42 | 0.42 | 0.36 |

with $RMSE_{xa}$ the root-mean-square-errors of the 1D-Var retrieved profiles with respect to the mast measurements, and $RMSE_{xb}$ the root-mean-square errors of the background profiles with respect to the mast measurements.

Table 3 gives a list of the different configurations. The CTRL run mimics the configuration of the operational AROME 3D-Var data assimilation system with a "climatological" **B** matrix taking into account cross-correlations between temperature and spe-
cific humidity. Config1 corresponds to the same configuration but removing the cross-correlations between temperature and specific humidity. Config2 mimics the use of a flow dependent **B** matrix with a full correlated fog-specific **B** matrix during fog events and a non-correlated climatological **B** matrix for all other weather conditions. For these three configurations, the bias-correction based on clear-sky O-B monitoring is applied to the raw BT measurements. Config3 is similar to config2 except that the bias-correction is not applied on the last four most opaque channels (54-58 GHz range). Config4 is similar to config3
except that the bias-correction applied to all channels is based on statistics of O-B departures made on clear-sky profiles with a temperature gradient between 500 m altitude and surface smaller than 5 K. Table 4 reports the calculated error reduction for each variable, each altitude and each 1D-Var configuration. The 1D-Var configuration maximizing each ER will be selected as the best configuration. Statistics are divided between fog profiles only (lower part) or all weather-conditions except fog (upper part).

The worst results are obtained with the CTRL configuration, which considers a "climatological" **B** matrix taking into account cross-correlations between temperature and humidity. With this configuration, the specific humidity RMSE with respect to tower measurements is degraded by up to 20 % (resp. 7 %) at 120 m altitude during fog conditions (resp. all weather conditions). This demonstrates the importance of the **B** matrix cross-correlations on 1D-Var accuracy and particulary in the case of observations with low information content on the vertical structure (as MWRs are mainly sensitive to the total column water
vapor content due to vertically quasi-constant weighting functions). The humidity profile degradation is significantly reduced to less than 3 % thanks to the use of a diagonal **B** matrix in Config2. Humidity profiles are finally improved by up to 21 % in



**Table 3.** List of 1D-Var experiments.

| Expt. | Description | Bias correction |
|---|---|---|
| CTRL | Climatological $\mathbf{B}_{clim}$ matrix computed from the AROME EDA with cross-covariances between T and Q | BC from AROME O-B monitoring |
| Config1: Bclim NO CROSS CORR | Climatological $\mathbf{B}_{clim}$ matrix computed from the AROME EDA without cross-covariances between T and Q | BC from AROME O-B monitoring |
| Config2: Bflow dependent | Cross-correlated $\mathbf{B}_{fog}$ matrix if visi_10m < 1000m $\mathbf{B}_{clim}$ without cross-correlations for visi_10m>1000m | BC from AROME O-B monitoring |
| Config3: Bflow dependent NO BC 54-58 GHz | Cross-correlated $\mathbf{B}_{fog}$ matrix if visi_10m < 1000m $\mathbf{B}_{clim}$ without cross-correlations for visi_10m>1000m | BC from AROME O-B monitoring for channels 22 GHz-53.86 GHz NO BC for channels 54.54 to 58 GHz |
| Config4: Bflow dependent BC $\delta T < 5K$ | Cross-correlated $\mathbf{B}_{fog}$ matrix if visi_10m < 1000m $\mathbf{B}_{clim}$ without cross-correlations for visi_10m>1000m | BC from AROME O-B monitoring base on all clear-sky profiles with $T_{500m}$-$T_{ground} < 5K$ |

**Table 4.** Error reduction (%) brought by the 1D-Var analysis over the background for all weather conditions (upper part) or only fog events (lower part). Statistics performed on temperature (T (K)) and specific humidity (Qspec, kg.kg$^{-1}$) at 50 and 120 m altitude.

| 1DVAR / ER | CTRL: $\mathbf{B}_{clim}$ Cross-corr | Config 1: $\mathbf{B}_{clim}$ no cross-corr | Config 2: $\mathbf{B}_{fog}$ $\mathbf{B}_{clim}$ | Config3: $\mathbf{B}_{fog}$ $\mathbf{B}_{clim}$ NO BC 56-58 GHz | Config4: $\mathbf{B}_{fog}$ $\mathbf{B}_{clim}$ BC $\Delta T < 5$ K |
|---|---|---|---|---|---|
| All conditions except fog (statistics on 2534 profiles) | | | | | |
| T 50m | 42 | 42 | 42 | 57 | 54 |
| T 120m | 40 | 40 | 40 | 50 | 50 |
| Qspec 50m | -4 | 0.2 | 0.3 | 0.3 | 0.3 |
| Qspec 120m | -7 | 0.1 | 0.1 | 0.1 | 0.1 |
| Fog cases (statistics on 351 profiles) | | | | | |
| T 50m | 37 | 37 | 34 | 50 | 44 |
| T 120m | 21 | 21 | 24 | 32 | 32 |
| Qspec 50m | -7 | -1 | -5 | 15 | 6 |
| Qspec 120m | -20 | -3 | 21 | 20 | 21 |

RMSE at 120 m during fog conditions with the use of a specific fog **B** matrix adapted to the meteorological conditions.

Table 4 finally shows that the best retrievals are obtained with Config3 and Config4 both in fog conditions and all-weather conditions. The improvement brought by these two configurations mainly comes from a change in the bias-correction applied to channels 11 to 14 (54-58 GHz). In fact, in Config3 these channels are not bias-corrected while in Config4 the bias-correction



was modified by removing atmospheric profiles during very stable conditions. This result demonstrates that, even though the bias-correction of MWR BT measurements can be computed from AROME short-term-forecasts for transparent channels, this method is not optimal for opaque channels without a thorough screening of the O-B innovations. In fact, the bias-correction of opaque channels depends on the accuracy of the forecast model within the boundary layer, which is known to be degraded during stable conditions. Similar conclusions are found in Martinet et al. (2017), despite the larger period of O-B monitoring (6-months instead of 2-months) and a less complex terrain.

In addition to comparison with tower measurements, temperature and humidity profiles have been evaluated through the whole atmospheric column in terms of bias and RMSE against 21 radiosondes (figure 3). Radiosondes were launched during atmospheric conditions prone to fog occurrence, mixing mainly stratus-cloud (cloud base below 1000 m) and fog conditions with a few profiles in clear-sky. 1D-Var retrievals are compared to the AROME background errors, which correspond to 1-hour forecast errors. For temperature, the best configuration shows RMSE smaller than 0.6 K within the fog layer and below 1.6 K considering the whole atmospheric profile up to 6 km altitude. The 1D-Var analysis outperforms the background in the first 800 m with a maximum improvement observed within the fog layer (RMSE decreased from 2.2 K to 0.6 K at 75 m). As expected, most of the information from the MWR observations are located below 2000 m and mainly below 1000 m. If the bias-correction based on the AROME monitoring is applied to the 54-58 GHz channels, a significant degradation in the temperature retrievals is observed in the first 500 m. The use of a "climatological" **B** matrix with cross-correlations degrades both temperature and humidity retrievals. The bias-correction has almot no impact on temperature retrievals.

For humidity, RMSE accuracies are less than 1 g.kg$^{-1}$ for the best scenario. Most of the improvement brought to the background is located below 3000 m with a maximum RMSE decrease reaching 0.2 g.kg$^{-1}$ at 75 m and 1800 m. 1D-Var retrievals on raw measurements without bias-correction only degrades slightly humidity retrievals by 0.15 g.kg$^{-1}$ above 2000 m. Overall, best performance of the 1D-Var is found with bias-correction applied on channels 1-10 and a dedicated fog **B**. This configuration is used in the following sections.

## 4 Thin radiative fog case study

This section focuses on a thin radiative fog case observed on the 28th of October 2016. Figure 4 shows the cloud based height retrieved from a CL31 ceilometer (top panel), the visibility measurements on the instrumented tower at 10 m and 120 m altitude (blue and green lines respectively, middle panel) as well as the 1-hour AROME forecasts of LWC for the same day (bottom panel). During the whole period, fog is only observed at 10 m altitude firstly during only 40 minutes at midnight and then during 4 hours from 5 to 9 UTC. A stratus-cloud is then observed from 10 UTC until midnight with a cloud base height between 300 and 500 m. The AROME backgrounds simulate a continuous thick fog event from 0 to 13 UTC, which is then lifted until 15 UTC into a stratus cloud at 500 m altitude. The stratus cloud is then dissipated to appear again after 20 UTC. In this example, two main deficiencies in the AROME 1-hour forecasts are observed: a temporally longer and vertically thicker fog event and the erroneous dissipation of the stratus cloud between 15 and 20 UTC.

Figure 5 compares the time series of temperature profiles (top panels) and specific humidity (bottom panels) forecast by

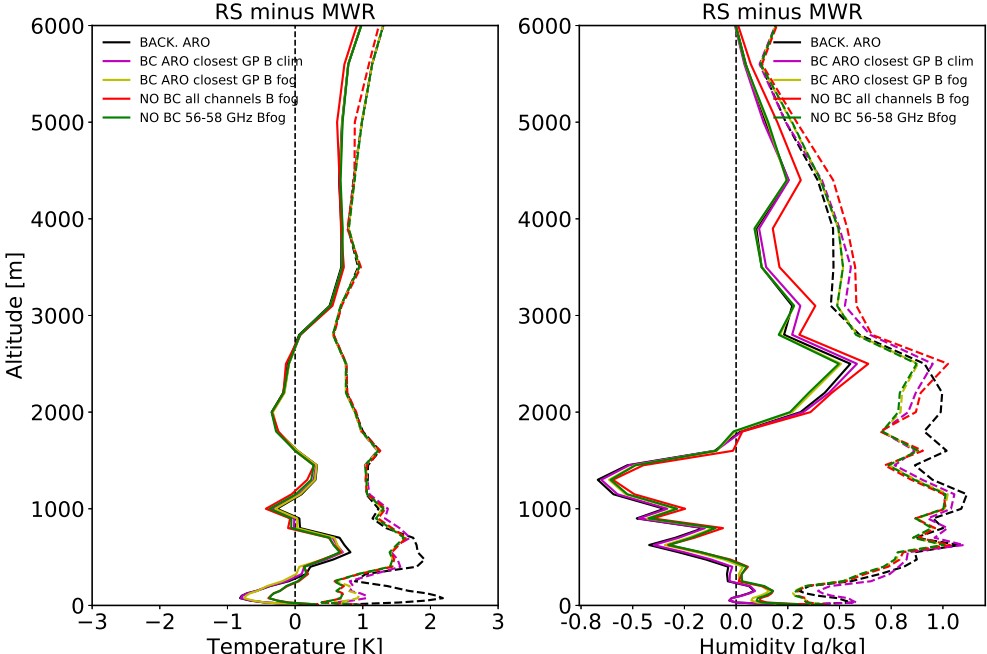

**Figure 3.** Vertical profiles of (left) temperature and (right) specific humidity bias (solid line) and root-mean-square errors (dashed lines) of 1D-Var retrievals (coloured lines) and AROME backgrounds (black line) against 21 radiosondes launched during IOPs: 1DVAR retrievals from AROME 1 h forecasts with bias-correction and a cross-correlated climatological **B** matrix (CTRL, magenta), with bias-correction and a cross-correlated dedicated fog **B** matrix (Config2, yellow), without bias-correction and a cross-correlated dedicated fog **B** matrix (red), with bias-correction except channels 11-14 and a cross-correlated dedicated fog **B** matrix (Config3, green).

AROME (left panels) and retrieved with the 1D-Var scheme either using the optimal configuration or the worst one. We can note the large temperature warming by up to 5 K from 0 to 12 UTC during the whole fog event (in the model space) in the 0-500 m vertical range. This is followed by a temperature cooling within 2 K during the stratus cloud (16 to 24 UTC). The worst configuration limits this warming during observed fog conditions through the whole fog thickness. The specific humidity is only modified during the fog event (5 to 9 UTC) with an increase of 0.1 g.kg$^{-1}$ in the first 1500 m. In the worst configuration the specific humidity is almost unchanged.

In order to quantify the accuracy of the 1D-Var increments in this specific fog case, figure 6 evaluates the corresponding diurnal evolution of temperature, specific humidity and relative humidity at 50 and 120 m altitude. A large underestimation of the temperature by 4 to 6 K is observed in the AROME forecasts by night until 13 UTC. AROME forecasts are also found to be too warm by 2 K after 18 UTC. The assimilation of MWR brightness temperatures in a 1D context greatly improves the



model background (temperature) during the nightime fog event with temperature errors smaller than 2 K after assimilation.
The 1D-Var retrievals almost perfectly fit the in-situ observations after 13 UTC for temperature both at 50 and 120 m altitude.
In terms of specific humidity, AROME tends to underestimate the specific humidity at nighttime probably due to an overestimation of the saturation. On the contrary, the specific humidity is overestimated in the afternoon. After 1D assimilation of MWR measurements, specific humidity is nearly identical to the AROME forecasts except during the longest fog event (between 4 and 9 UTC) where the 1D analysis is closer to the tower measurements than the background. This is due to the use of the cross-correlated fog **B** matrix under these conditions. Most of the model increment is thus produced by the **B** matrix cross-covariances. Background errors are reduced from 0.5 g.kg$^{-1}$ to 0.1 g.kg$^{-1}$. In terms of relative humidity, the temperature warming by night manage to un-saturate the fog layer in agreement with the tower in-situ measurements. However, this field is degraded after 13 UTC. In fact, the 1D-Var scheme correctly reduces the temperature but is not able to decrease the specific humidity. The relative humidity is thus wrongly increased by the 1D-Var analysis.

In view of the future inclusion of hydrometeors in the data assimilation control variables, the information brought by MWRs to the liquid water path (LWP) could also be very valuable. Figure 7 shows the time series of LWP forecast by AROME, retrieved through the 1D-Var and retrieved from a quadratic regression applied on BT measurements. It can be seen that the AROME model clearly overestimates the fog LWP with a maximum reaching 90 g.m$^{-2}$ at 7 UTC decreased down to 25 g.m$^{-2}$ after the 1D assimilation of MWR brightness temperatures. During the missing stratus cloud, the LWP is significantly increased with values between 30 and 80 g.m$^{-2}$ even if the background profile has no cloud layer between 14 and 20 UTC. These LWP modifications brought by the 1D-Var are consistent with the in-situ observations on the instrumented tower as well as ceilometer observations.

## 5 Six-month statistics

While previous section focuses on one extreme fog case, this section aims at more general conclusions on the expected impact of MWR BTs assimilation on AROME analysis. To that end, a statistical evaluation of the expected model increments (analysis minus background differences) after assimilating MWR measurements has been conducted using the tower measurements during the six-month period. 1D-Var retrievals have been performed using the optimal configuration described in section 3.2. A total of 351 hours of fog (rain events have been removed) could be observed with the MWR. In order to evaluate the performance of the AROME background profiles (1h forecast) to accurately forecast fog events, statistics based on the hit ratio (HR), false alarm rate (FAR), frequency bias index (FBI) and critical-success-index (CSI) have been computed. If GD is the number of fog profiles well detected, ND the number of undetected fog profiles, FA the number of false alarms, these scores are defined by:





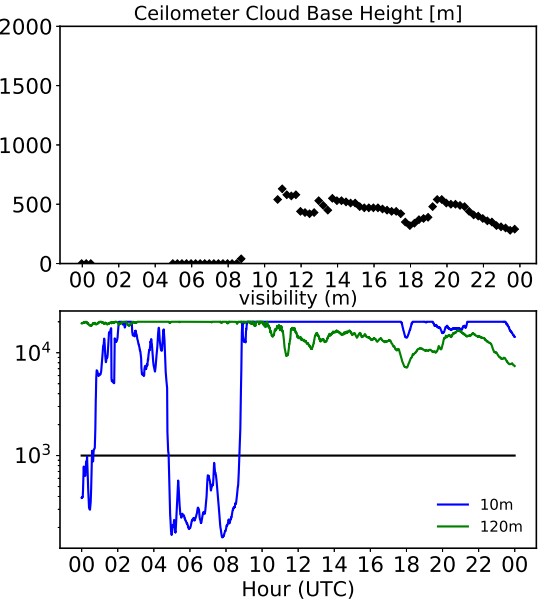

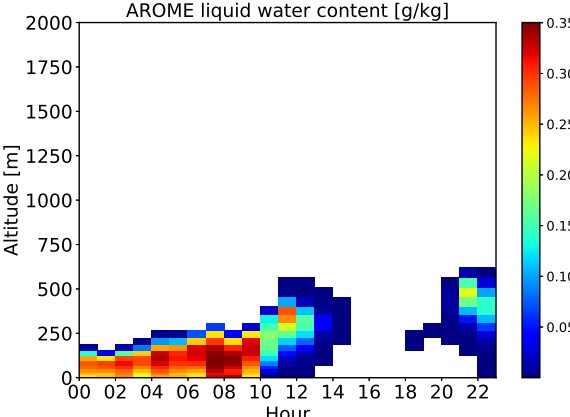

**Figure 4.** Top panel: Cloud base height (m) derived from the CL31 ceilometer measurements, middle panel: visibility at 10 m (blue) and 120 m (green line), bottom panel: AROME 1-hour forecasts of liquid water content in g.kg$^{-1}$. 28 October 2016.



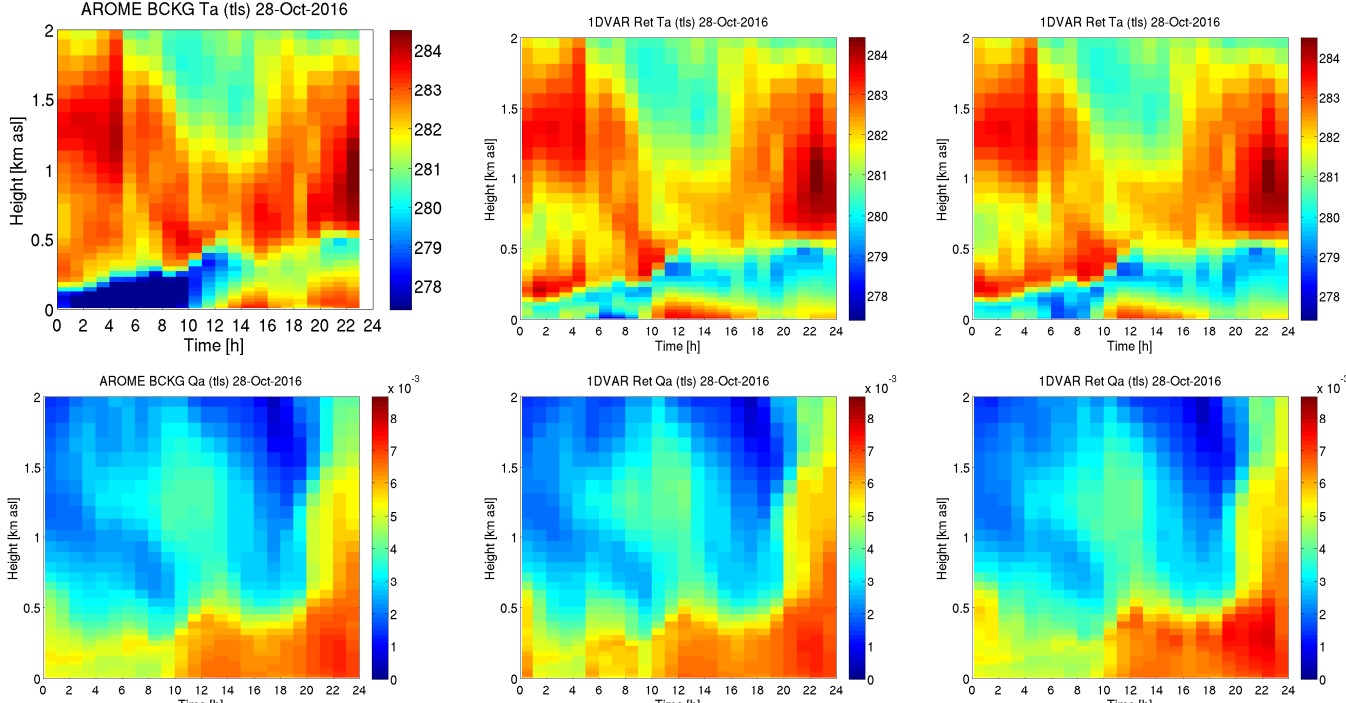

**Figure 5.** Time series of temperature profiles (top panels) and specific humidity (bottom panels) forecast by AROME (left panels) and retrieved with the 1D-Var scheme either using the optimal configuration (Config3, middle panels) or the worst one (CTRL, right panels). 28 October 2016.

$$
\begin{aligned}
HR &= \frac{GD}{GD+ND} \\
FAR &= \frac{FA}{GD+FA} \\
FBI &= \frac{GD+FA}{GD+ND} \\
CSI &= \frac{GD}{GD+ND+FA}
\end{aligned}
\tag{1}
$$

To detect fog profiles in the model space, a new visibility diagnosis specifically developed for the AROME model has been
used (Dombrowski-Etchevers et al. (2020)). A hit ratio of 73 % and a false alarme rate of 58 % was found. A FBI of 1.77
means that the AROME background profiles tend to forecast too many fog events. CSI equal to 0.35 means that only 35 % of
fog events (observed and/or predicted) are correctly forecast by the model. These statistics emphasize that quite large errors are
observed in the AROME 1h forecasts of fog with an excessive number of false alarms. In order to evaluate the potential benefit
of MWR observations to adjust the AROME background profiles, the statistical study of model increments is split between the
good detections, missed fog profiles and false alarms.





Firstly, the frequency distributions of differences of 1D-Var analysis and background with tower measurements at 50 m are displayed in figure 8 both for temperature and specific humidity. For temperature and for all subsets, the distributions of 1D-Var analysis errors are more centered and more symmetric compared to the background error distributions. Thus, the largest background errors (above 2 K in absolute values) are succesfully corrected by the 1D-Var analysis. Background error

distributions also present a larger tail towards negative values with a secondary peak centered around -4 K in the case of false alarms and to a smaller extent in the case of good fog detections. The largest temperature improvement is observed in the case of false alarms with only 35 % of the background errors being within -0.5 to 0.5 K, against 69 % for the analysis. RMSE with respect to tower measurements are also significantly improved with values between 1.3 and 1.9 K in the background against 0.6 K in the analysis. The frequency distribution of specific humidity errors for 1D-Var analysis and background are close, with

similar bias and RMSE for good detections and false alarms. A slight degradation is observed for missed fog detections with a RMSE of 0.33 g.kg$^{-1}$ in the analysis against 0.25 g.kg$^{-1}$ in the background. Overall, the impact on humidity is less evident than on temperature at 50 m altitude.

To get a vertical perspective, figure 9 shows the profiles of the frequency distribution of analysis minus background differences. As more than 90 % of the water vapour increments are within 1 g.kg$^{-1}$ up to 1500 m altitude, only the impact on temperature

is discussed. For each vertical bin, the frequency of the temperature increments within a given range of values is shown. The frequency distribution of 1D-Var increments has been separated between cases of correct fog detection, missed fog and false alarms. For all dataset, most of the temperature analysis increments are observed below 750 m and span the range -5 to 5 K. The largest increments are observed between 100 and 300 m altitude for which around 20 % of the analysis minus background differences are larger than 2 K in absolute values. We can note significant differences in the shape of the increment distributions

depending on the forecast score. While the distribution of good detections is quite symmetric, it is not the case for missed fog profiles and false alarm distributions. In the case of missed fog events, the distribution is negatively-skewed close to the ground whereas it is positively-skewed above 100 m altitude. This asymmetry means that the largest analysis increments in magnitude tend to decrease the temperature close to the ground and increase the temperature above 100 m. Consequently, we can expect 1D-Var analyses to increase the atmospheric stability in the first 150 m, which is key for fog formation. In the case of false

alarms, the distribution is positively-skewed for all vertical levels. This asymmetry means that the largest analysis increments, though less frequent in the distribution, occur when the AROME forecasts tend to significantly overestimate the temperature cooling. By limiting the temperature cooling, the 1D-Var analyses might limit the erroneous saturation leading to false alarms in the background.

Additional value of MWR data for NWP forecasts and process studies is in the LWP product. In fact, MWR is one of the

most reliable sources for this variable (Crewell and Löhnert (2003)), which is key for better understanding the microphysics of fog lifecycle and limiting the forecast spin-up (i.e. the unbalance of thermodynamic profiles with microphysical variables during the analysis). As hydrometeors are currently not included in most operational variational data assimilation schemes, the background profiles used in this study correspond, in fact, to the analyzed fields, thus to the profiles used to start the new forecast every hour. The statistical study performed here is also useful to evaluate the expected impact on the AROME analyses

if MWR observations were assimilated and the LWP included in the control variables. To that end, figure 10 investigates the





frequency distribution of LWP increments split by forecast skill (good detections, undetected fog, false alarms). Firstly, we can note that the LWP increments are higher than 50 g.m$^{-2}$ in absolute values for approximately 50 % of good detections and missed fog profiles and 30 % of false alarms. During false alarms, 95 % of the background LWP values are below 20 g.m$^{-2}$ (not shown), which is close to the MWR sensitivity which might explain smaller 1D-Var increments during false alarms. The

mean increment is the highest in the case of missed fog events (57 g.m$^{-2}$) and the smallest in the case of false alarms (15 g.m$^{-2}$). It is important to note that during false alarms, the LWP increment might be positive due to the presence of cloud layers though we would expect the 1D-Var analysis to decrease the LWP within the fog layer. If we restrict the statistics to false alarms without cloud aloft, the mean increment is reduced to -2 g.m$^{-2}$. As expected, large positive increments occur more often in fog cases un-detected by AROME with 47 % of the distribution showing increments above 50 g.m$^{-2}$ against

35 % in good detections and 22 % in false alarms (8 % for false alarms without cloud layers aloft). To further investigate the LWP increments and retrieved values, more in situ data are necessary, e.g from the cloud droplet probe mounted on the tethered balloon or cloud radar measurements. However, the lack of cloud radar measurements to differentiate the LWP within the fog layer and cloud aloft makes this evaluation complex. Too few cases during which MWR observations were colocated with an entire sounding of the fog layer with the tethered balloon have been sampled to make an independent evaluation of

this product. This is why we use the LWP derived from the MWR alone through a quadratic regression as a reference. The expected accuracy of this product 15 to 20 g.m$^{-2}$ according to Crewell and Löhnert (2003). To that end, figure 12 shows the scatterplot between the LWP retrieved with the MWR alone (through multi-channel regressions provided by the manufacturer) and the 1D-Var analyses or background profiles (left panel). We can note the large improvement in correlation between the LWP forecast by the background (0.72) versus the 1D-Var analysis (0.98) with respect to the MWR multi-channel retrieval.

This is of course expected as the 1D-Var minimization tends to get closer to the MWR brightness temperatures which are also used in the multi-channel retrieval. However, this evaluation is a good sanity check showing the good behaviour of the 1D-Var algorithm and its capability to extract the information from the observation even with very large errors in the first guess background profiles. The mean error of the AROME LWP is -49 g.m$^{-2}$ and is reduced to -2 g.m$^{-2}$ after 1D assimilation. The root-mean-square error is significantly reduced from 102 g.m$^{-2}$ to 27 g.m$^{-2}$.

The same evaluation has been carried out on the IWV (figures 11 and 12). Since MWRs are more sensitive to column integral than vertical distribution, more significant impact is expected on IWV than specific humidity profiles. The IWV increments span from -4 to 4 kg.m$^{-2}$, which correspond to a change in the background IWV up to 30 %. The distribution of IWV increments is positively-skewed for correct fog detection meaning that the largest increments in magnitude are observed when the background underestimates the integrated water vapour content. On the contrary, it is negatively-skewed for missed fog

profiles meaning that the largest increments occur when the model overestimates the integrated water vapour content. It is more symmetric in the case of false alarms. The correlation coefficient with respect to the MWR multi-channel retrieval (fig. 12) is sligthly increased from 0.97 to 1. The RMSE is improved from 1.30 to 0.71 kg.m$^{-2}$. The impact of MWR observations is thus positive on IWV, though the good quality of AROME humidity forecast leaves little room for improvement. This could be explained by the assimilation of observations sensitive to the total column water vapor like Global Navigation Satellite System

(GNSS) zenith total delay. Further investigation on multiple sites would be needed to confirm this hypothesis.





## 6  A regional-scale MWR network for fog process studies: the SOFOG3D experiment

This worked has proved MWRs to be potential good candidates to be assimilated into current mesoscale models with a special focus on fog forecast improvement. However, our conclusions are currently limited by the small dataset (only one winter at one site) and the lack of impact studies on fog forecast. Although, a positive impact is expected on the analysis of the
ABL temperature profile and the LWP, and, to a smaller extent, to the IWV, the next step will be to quantify the impact of a more accurate initial state on fog forecast capability. Among the massive number of observations currently assimilated into operational models, the assimilation of only one MWR unit would probably not be efficient to effectively constrain the boundary layer in the model analysis and to keep the valuable information brought by this local observation over the forecast range. In order to go further in this evaluation, the deployment of a dense network of MWRs is necessary to perform a data
assimilation study into the operational AROME 3D-Var assimilation system. Thanks to the strong European collaboration built in the framework of the cost action TOPROF (www.cost.eu/COST_Actions/essem/ES1303), pursued by the cost action PROBE (PROfiling the Boundary layer at a European Scale, Cimini et al. (2020)), an un-precedented regional-scale network of 8 MWR units has been deployed in the South West of France during the period October 2019 to April 2020. This network will serve the data assimilation experiment, fog process studies and model evaluation of the international SOFOG3D (SOuth
FOGs 3D experiment for fog processes study) experiment led by Météo-France. Figure 13 shows the domain of the dedicated 500 m horizontal resolution AROME version in test for evaluation during SOFOG3D and the location of the 8 MWR units deployed for the experiment. The super-site is indicated by the white hatched area. MWR locations have been chosen for an homogeneous spread over the AROME domain at sites known for the high frequency of fog occurrence. An increased density of MWRs is found at the super-site with two co-located MWRs and a third humidity profiler deployed approximately 7 km
away from the super-site to document the impact of surface heterogeneities on fog characteristics. The methodology introduced in this paper will be extended to the 8 MWRs deployed during SOFOG3D. This large dataset will help quantifying the spatio-temporal variability of fog parameters (thermodynamics and microphysics) between the different sites, better understand the main processes playing a role in fog formation / dissipation / development and run real data assimilation experiments using the operational 3D-Var assimilation scheme of the AROME model to quantify the expected fog forecast improvement thanks to
ground-based MWRs.

## 7  Conclusions

In this study, the expected benefit of ground-based MWRs on NWP analyses during fog conditions has been investigated thanks to a 1D-Var technique. Temperature, humidity and LWP have been retrieved through the optimal combination of short-term-forecasts and MWRs brightness temperatures. In this study, a new retrieval algorithm, combining the NWPSAF 1D-Var and
the fast radiative transfer model RTTOV-gb, has been evaluated on a 6-month period spanning 351 hours of fog conditions. The first part of this work aimed at deriving an optimal background-error-covariance matrix for fog conditions with the use of newly developed AROME EDA. Similarly to Ménétrier and Montmerle (2011), background-error standard deviations were found to be approximately 40 % larger within the first 250 m for temperature compared to a commonly used "climatological





**B** matrix". For specific humidity, similar standard deviations were observed. Most of the differences between a climatological

**B** matrix and a fog **B** matrix were observed in the cross-correlations between temperature and specific humidity, with a strong positive coupling within the fog layer and uncoupling between the fog layer and atmospheric layers above. The impact of the **B** matrix and bias-correction has been investigated through a statistical evaluation of the retrieval accuracy with respect to the in-situ measurements on the instrumented tower at 50 and 120 m altitude. The optimal configuration has been defined through the definition of the error reduction brought by the analysis over the background for each variable (temperature and specific

humidity) and each altitude. The best scenario mimics the use of a "flow dependent" **B** matrix by using a cross-correlated fog **B** matrix when fog is detected by visibility measurements and an un-correlated climatological **B** matrix during the other conditions. The retrievals of specific humidity at 120 m altitude are the most impacted: contrary to the significant degradation of the background by around 20 % with a sub-optimal **B** matrix, an improvement of 21% of the background is obtained with an optimal **B** matrix. This demonstrates the crucial role of the B matrix cross-correlations when assimilating observations with low

information content on the vertical structure. Consequently, the on-going development of an 3D-EnVar scheme for the AROME model (Montmerle et al. (2018)) is a necessary step to optimally assimilate MWR observations into the AROME model. The use of a static bias-correction based on the monitoring of observation minus background innovations was also evaluated. Biases of less than 0.5 K were observed for K-band and opaque V-band channels and up to -4.7 K for the most transparent V-band channels. The found bias is similar to previous studies; its correction applied to BT measurements improves humidity retrievals

above 2000 m but degrades temperature retrievals in the first 200 m. This degradation is most likely due to well-known larger model errors in the boundary layer during stable conditions, which are incorrectly included in the bias-correction. Restricting the computation of the bias-correction to clear-sky unstable conditions was found to remove most of the degradation. Overall, with the best configuration (flow dependent fog **B** matrix and no bias correction for most opaque channels), temperature and humidity profiles could be retrieved with RMSE below 1.6 K and 1 g.kg$^{-1}$ up to 6 km in the troposphere.

A thin radiative fog sampled during the first IOP of the experiment was then described. For this specific case, the AROME model was found to simulate a temporally longer and vertically thicker fog event and is not able to maintain the stratus cloud in the afternoon. After 1D assimilation of MWR observations, a large warming up to 5 K is observed within the first 500 m during the fog event associated with an increase in specific humidity and a decrease of LWP by 40 to 70 g.m$^{-2}$ consistent with in-situ measurements showing the large impact brought by MWR observations to modify the initial state of the model in fog

conditions.

Finally, a statistical evaluation of the expected model increments after assimilating MWR measurements has been conducted using tower measurements. Large forecast errors were observed in the AROME backgrounds with a tendency to overestimate the presence of fog. During missed fog profiles, 1D-Var increments pull towards lower temperature close to the ground and higher temperature above 100 m altitude, i.e. higher atmospheric stability. The largest analysis increments and background

errors are observed during false alarms when the AROME forecasts tend to significantly overestimate the temperature cooling. Overall, RMSE values from 1.3 to 1.9 K are observed in the background against 0.6 K in the analysis. For specific humidity, analysis increments are small and below 1g.kg$^{-1}$ within the fog layer. On the contrary, a large impact has been found on the LWP with increments up to 200 g.m$^{-2}$ in extreme missed fog events. A larger impact was found on the IWV than the


humidity profile with a RMSE with respect to tower measurements decreased from 1.3 kg.m$^{-2}$ to 0.7 kg.m$^{-2}$ during observed
fog profiles. However, it was noted that the AROME backgrounds are more accurate on the IWV compared to temperature and
LWP, which leaves less chances for improvement.

Using for the first time the RTTOV-gb fast radiative transfer model, this study investigated the impact of assimilating MWR
observations in the AROME model during fog conditions. This evaluation, previously limited to temperature profiles only, was
extended to humidity and LWP. Promising results are shown, with significant positive impact on temperature and LWP, while
small but slightly positive impact on humidity. In order to confirm the results obtained in a 1D-Var framework, the next step is
now to assimilate a real network of ground-based MWRs through a 3D-Var or 3D-EnVar data assimilation scheme. Following
the recommendations of Caumont et al. (2016) and thanks the strong European collaboration built within the TOPROF and
PROBE COST actions, 8 MWRs have been deployed in the South-West of France from October 2019 to April 2020 in the
context of the international fog campaign SOFOG3D. The locations of the MWR units have been chosen to optimize their
impact in the model specifically for fog forecast evaluation. A 1D-Var plus 3D-EnVar approach will be used to assimilate
profiles retrieved through the 1D-Var algorithm presented here, taking the most out of the lessons learnt in this work.

*Data availability.* The AROME forecasts are available on request on the website https://donneespubliques.meteofrance.fr/. Instrumental data
are available on request to: frederic.burnet@meteo.fr.

*Author contributions.* PM supervised the MWR deployment during the field experiment, processed all the data, led the scientific analysis
and wrote the paper. DC participated to the development of the 1D-Var algorithm and scientific analysis of the results. FB supervised the Bure
experiment and participated to the scientific analysis of the results. VU was in charge of the technical deployment of the MWR during the
experiment. BM developed and provided the support for the software used to derive background-error-covariance matrices from ensemble
data assimilation. YM provided the AROME ensemble data assimilation outputs to compute the background-error-covariance matrix and
participated to the redaction of section 3.1.

*Competing interests.* The authors declare that they have no conflict of interest.

*Acknowledgements.* This article is based upon work from COST Actions ES1303 (TOPROF) and CA18235 (PROBE), supported by COST
(European Cooperation in Science and Technology) – www.cost.eu. The authors thank ANDRA for providing access to the atmospheric
platform observations. The deployment of the SOFOG3D MWR network was funded by the ANR SOFOG3D (SOuth west FOGs 3D
experiment for processes study, ANR-18-CE01-0004). Thomas Rieutord is thanked for helpul discussions on statistical analysis. Yann Seity
is thanked for providing the map of the AROME 500m domain.



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





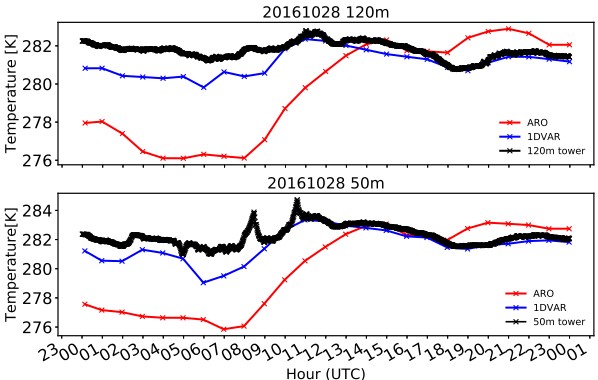

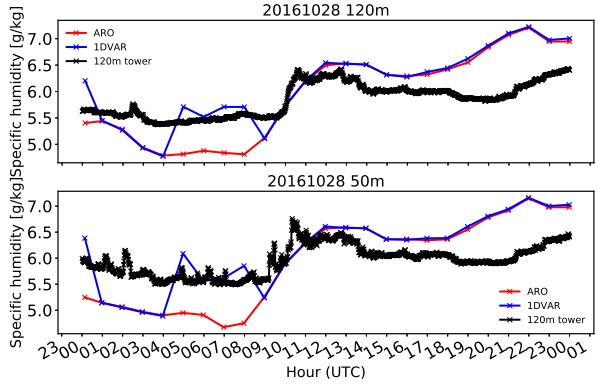

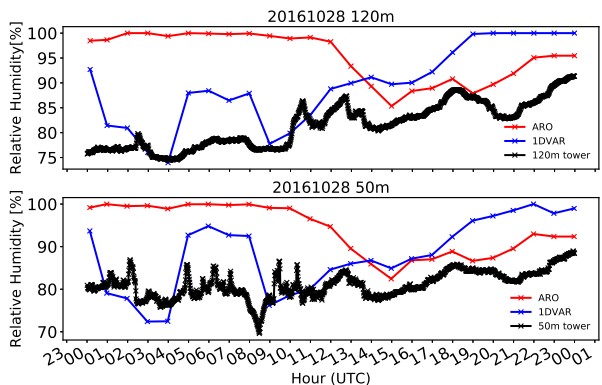

**Figure 6.** Diurnal evolution of temperature (top panel), specific humidity (middle panel) and relative humidity (bottom panel) forecast by AROME (red), measured by weather station (black) and retrieved by the 1D-Var algorithm (blue). 28 October 2016.

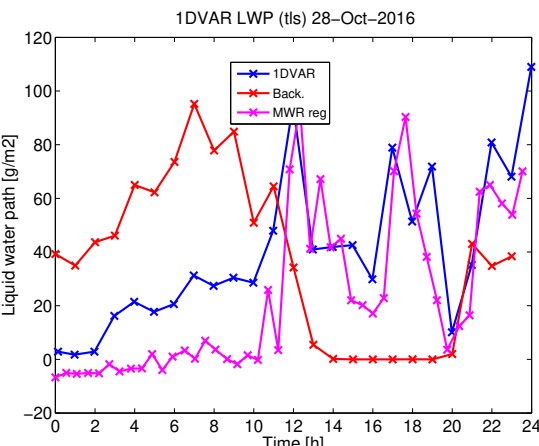

**Figure 7.** Time serie of liquid water path forecast by AROME (red), retrieved by the 1D-Var algorithm (blue) or retrieved from the MWR alone through a quadratic regression (magenta). 28 October 2016.





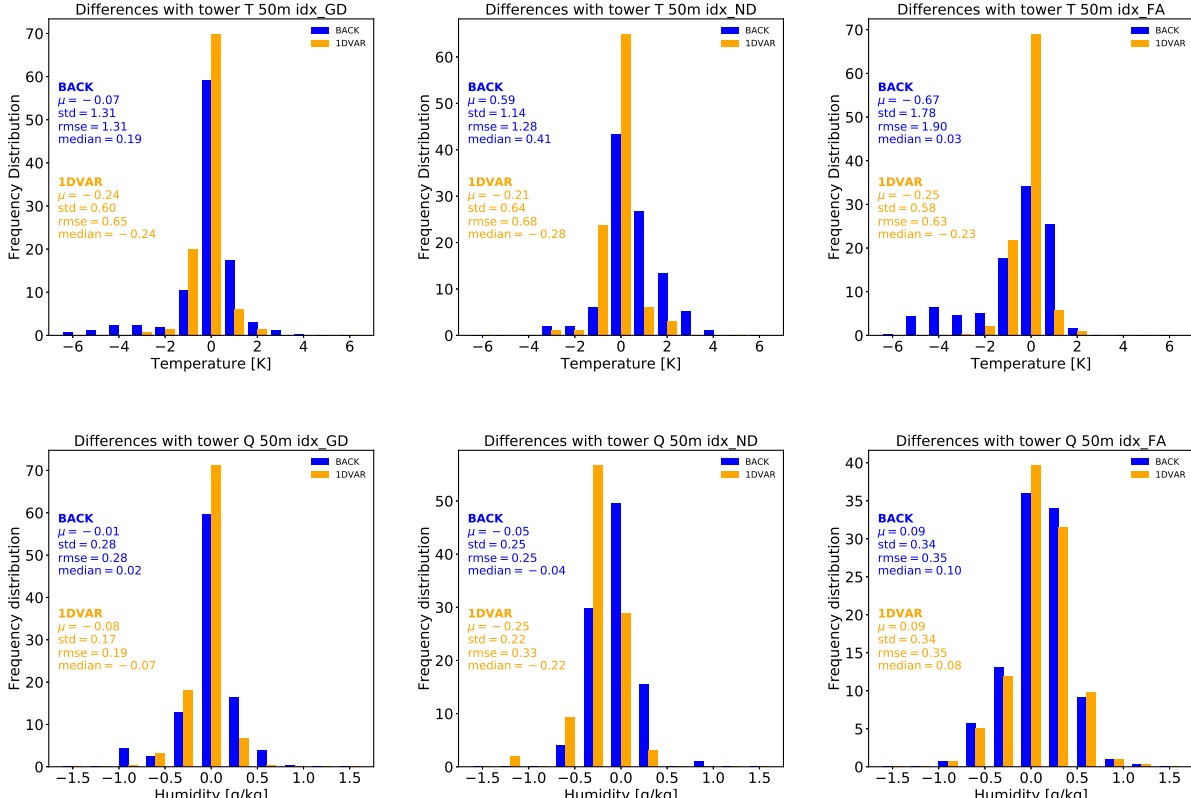

**Figure 8.** Frequency distribution of 1D-Var analyses (orange) and background (blue) differences with tower measurements for temperature (top panel) and specific humidity (bottom panel) at 50 m altitude. Statistics performed over 255 profiles of good fog detection (left panel), 95 profiles of undetected fog (middle panel) and 368 profiles of false alarms (right panel).

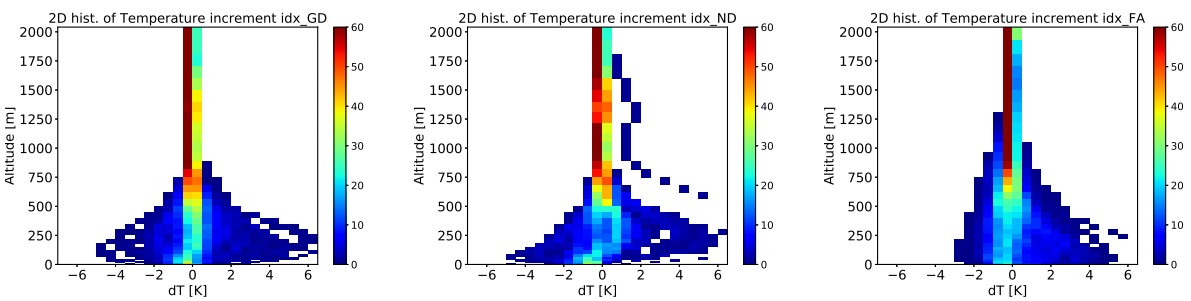

**Figure 9.** Vertical profiles of the frequency distribution of temperature increments (analysis minus background differences). Statistics performed over 255 profiles of good fog detection (left panel), 95 profiles of undetected fog (middle panel) and 368 profiles of false alarms (right panel).





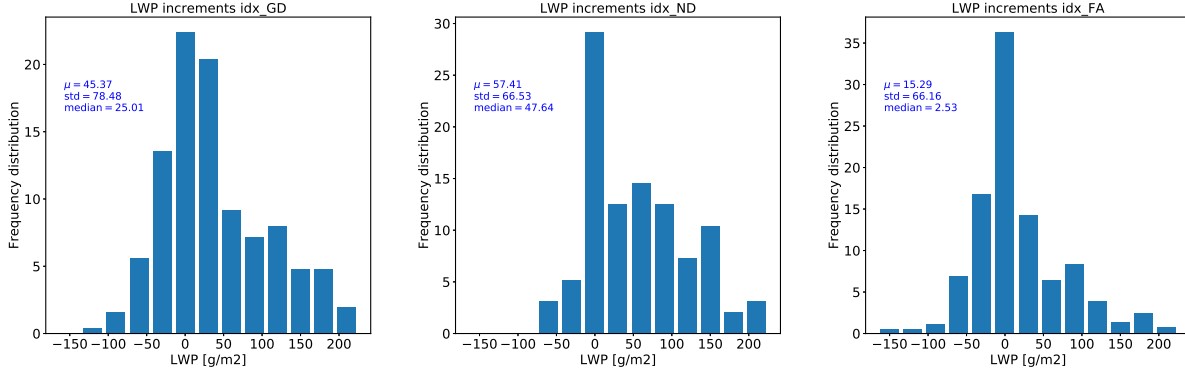

**Figure 10.** Frequency distribution of 1D-Var LWP increments (g.m$^{-2}$). Statistics performed over 255 profiles of good fog detection (left panel), 95 profiles of undetected fog (middle panel) and 368 profiles of false alarms (right panel).

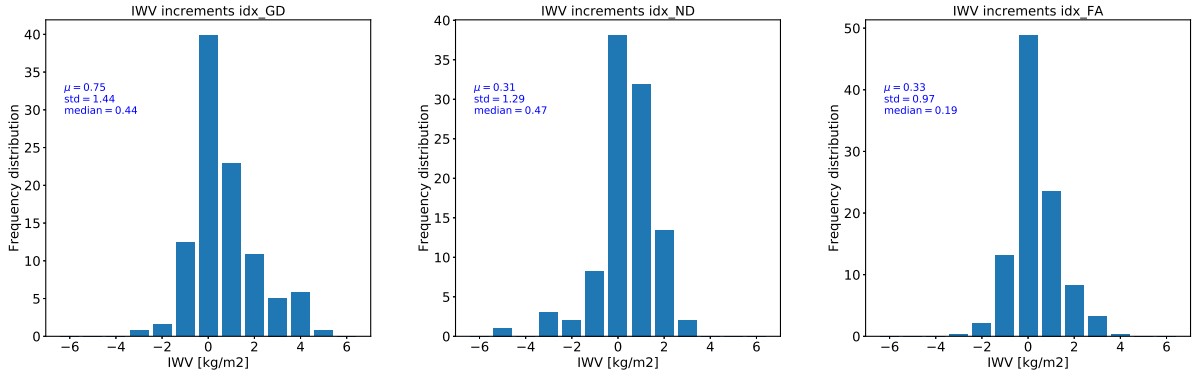

**Figure 11.** Frequency distribution of 1D-Var IWV increments (kg.m$^{-2}$). Statistics performed over 255 profiles of good fog detection (left panel), 95 profiles of undetected fog (middle panel) and 368 profiles of false alarms (right panel).





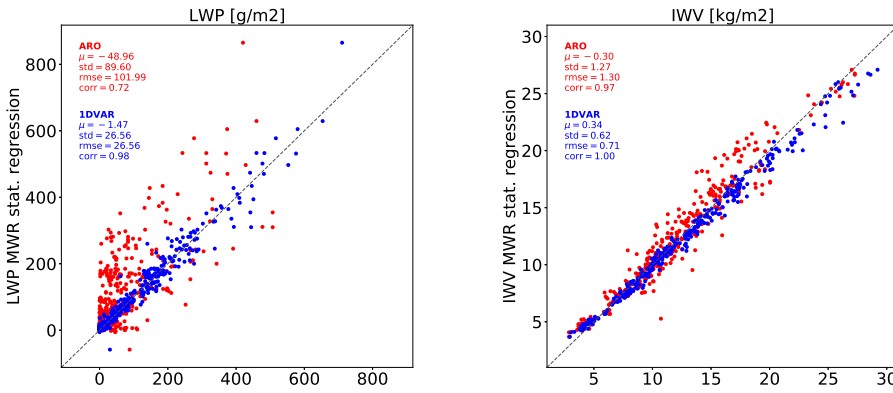

**Figure 12.** Scatterplot between a multi-channel regression based on MWR observations (y-axis) and the background forecast by AROME (red dots) or the 1D-Var analysis (blue dots) for LWP (left panel) and IWV (right panel). Statistics performed over 351 observed fog profiles.

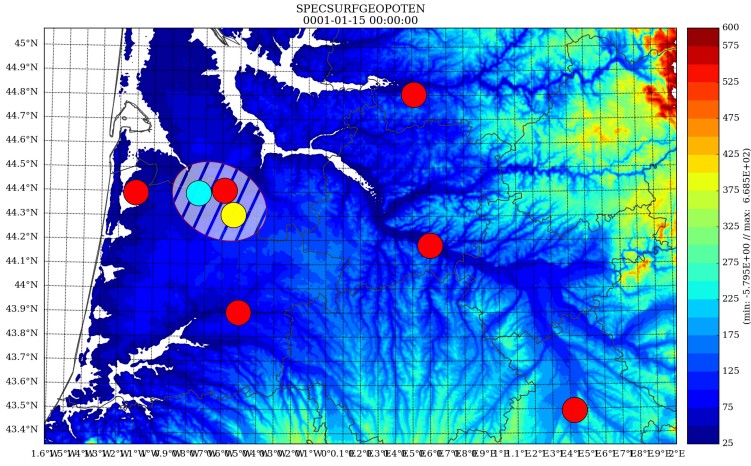

**Figure 13.** Surface geopotential and domain of the AROME-500m dedicated to the SOFOG3D experiment. Locations of the MWR sites are shown with the filled circles (red indicate temperature and humidity profilers, yellow only humidity retrievals and cyan only temperature retrievals. )