# Peer review of "Improvement of numerical weather prediction model analysis during fog conditions through the assimilation of ground-based microwave radiometer observations: a 1D-Var study."

_Atmospheric Measurement Techniques, 2020_

## Referee Comment (RC1) · Anonymous Referee #1 · 10 Jun 2020

General comments

The manuscript describes the assimilation of brightness temperatures measured by a ground-based microwave radiometer under fog conditions with a 1D-Var approach. It is shown how the profiles of temperature and humidity are corrected in the analysis. The manuscript is well written (structure, language) and comprehensible with illustrative figures. The topic is highly relevant as a first step towards the assimilation of the MWR

data in numerical weather prediction models and the findings of the study give valuable hints how to assimilate this kind of data successfully with respect to the background covariance matrix and bias correction and what kind of increments are to be expected on the profiles. The content of the manuscript is closely related to the measurements and thus still fits the scope of AMT, however a follow-up paper focusing more on the impact of the data on model forecasts might fit better to another journal. In some paragraphs of the manuscript, however, the explanations are kept very short, here the manuscript could benefit in terms of comprehensibility, also for a broader readership, if the statements would be explained a bit better. Thus I suggest the manuscript for publication with some minor corrections and clarifications which I list below.

Specific comments

Abstract: The authors should mention already in the abstract that the brightness temperatures are assimilated directly over a forward operator (RTTOVgb)

Line 31-33: This formulation is not exact: it is true that satellite data provide limited information on the ABL, but not because of the complexity of data assimilation over lands, to be exact this issue makes the use of the data for NWP more difficult. Please rephrase.

Line 44: it could be emphasized here that the study by Otkin and Hartung et al. (2011) with 140 MWRs was an OSSE ( in contrast to your study using real data)

Line 123: For clarification for readers not familiar with (MW) remote sensing you could add half a sentence why transparent channels are omitted at low elevation angles

Line 134: The authors could add a sentence, why it is inadequate for fog areas

Line 150-151: This is not clear. What kind of tests?

Section 3.2: This section is not structured well. First it is about obs errors, then bias correction, then obs errors, then both. . .

[Figure]

Line 176: what do the authors mean by the "individual errors which were added in quadrature"? Not really clear to me.

Line 214-215: not fully clear... So the authors want to say the dataset consists of stratus clouds, profiles with fog, and some clear-sky profiles?

Line 251-256: Please give more explanations in this paragraph, why the underestimation of specific humidity at nighttime is due to an overestimation of saturation, and why most of the model increments are produced by the B matrix cross-covariances.

L264: For clarification his could be rephrased to "... During the period where the model fails to simulate the stratus cloud, the LWP is significantly increased in the 1D analysis with values between..."

L279: The authors could add one sentence on what the visibility diagnosis is based.

L317-318: This is not clear. Does it mean the profiles used are not forecasts but taken from an analysis with conventional data already assimilated?

Technical corrections

L31: Better: "... wich is undersampled by observations. "Even though satellite data provide a global coverage..."

L48: Better: impact of this network was found to be neutral.."

L51: Better: "AROME model with a one-dimensional.."

L55: correct to: "... and evaluates the impact...."

L122: correct to: "... consists of.."

L129: replace "spatially" by "horizontally" (because spatially comprises vertical and horizontal directions)

L144: Better: "...with a horizontal resolution set to 3.2km and ..." ("finally" should be omitted)

[Figure]

L167: no comma here

L167: better: "but also on an adequate specification of. . ."

Line 201: do you mean Config1 here?

General: References to figures in the text should be with capital "F". "Figure X" instead of "figure X".

L222: Typo: "almost"

L229: Typo: cloud base height

L232: better: ". . . fog is observed at 10m altitude during 40 minutes at midnight and. . ."

L257: better: ". . . by night leads to the effect that the fog layer is not saturated any more in agreement. . ."

L262-64: Better: ". . . with a maximum reaching 90gm-2 at 7UTC. This value, however, decreased down to. . ."

L269: Better: "While the previous focuses on an extreme. . ."

L382: Better: ". . . has been investigated with. . ."

L429: Better: ". . . on temperature and LWP and small but. . ."

Figure 8 Caption: should be re-phrased to: ". . . differences compared to tower measurements. . ."

Figure 13: The axes are difficult to read. Maybe the figure could be enlarged to improve this.

[Figure]

---

## Referee Comment (RC2) · Anonymous Referee #2 · 17 Jun 2020

Comments on "Improvement of numerical weather prediction model analysis during fog conditions through the assimilation of ground-based microwave radiometer observations: a 1D-Var study" by P. Martinet et al.

**Summary:**

The paper examines the assimilation of ground-based microwave radiometer observations into a Numerical Weather Prediction model and concludes that radiometers can benefit the forecast of fog events by reducing the temperature errors in the model.
I found the work interesting and generally understandable, I have only a few comments on the discussion of the background covariance and a few clarifications that may be needed to make the paper more accessible to a general audience.

**Major comment:**

I understand the discussion about optimal estimation in section 2.3 and 3.1, however I am a little bit perplexed by the discussion of the background covariance in section 3.2.

From what I could understand the a priori vector **Xb** = (**T**, **Q**, LWP) used in the convergence scheme specified in line 110 is provided by 1 hr AROME forecast profiles.

The corresponding background covariance **B** associated with this a priori estimate was estimated as described in line 145-150 for all cases and for a subset of fog cases and the diagonal terms were multiplied by 0.7. Where I get lost is the next section (3.2) where the background covariance is modified in a seemingly arbitrary fashion by removing the cross-correlation terms from the climatology. I entirely understand using a fog covariance for the fog cases and a climatology covariance for the non-fog case. However, to "choose" the background covariance that optimizes the retrieval results seems a little bit unorthodox.
My point is that the covariance should be an "objective" way (as far as possible) to quantify the uncertainty associated with the a priori information. It seems that if the background covariance associated with the **Xb** is not good enough for the retrieval perhaps a different choice for the a priori **Xb** should be made (i.e. not from the model but perhaps from a radiosonde ensemble).

Alternatively, the convergence could be controlled with a multiplicative factor to **B** (usually called $\gamma$) that is reduced at each iteration based on the behavior of the cost function. This approach is mostly used for infrared retrievals, but, in this case, it may prove beneficial as well. *So perhaps I am not entirely understanding this part, in which case this procedure of "choosing" **B** based on the retrieval results could be better justified or may be the straight optimal estimation approach should be modified the way mentioned above.*

**Minor comments:**

Abstract: There is terminology that is not defined for example, in lines 11, 12, 15, what are 1D-var increments? *I suggest either making the abstract less detailed about the results or defining the terms used.*

Table 4 is not clear. The caption says "Error reduction (%)" over the background. It is not clear what the negative number means. *Does it mean that the retrieval is actually increasing the RMSE with respect to the background?*

Table 4: Just to make sure I understand correctly, this statistic is computed by taking one layer of the retrieval profile grid corresponding to the tower height of 50 and one layer corresponding to 120 m? I would imagine that the retrievals at these two layers are highly correlated because the vertical resolution of the radiometer at this height is about 100 m. *Therefore, is there even a merit to look at two layers vs. averaging the tower and radiometer measurements between 50 and 120m?*

Fig. 3 It is not clear what "closest GP B clim" mean in the labels.

In section 4 line 203 is said that "*the best retrievals are with config 3 and config 4*" and later at line 225 it is said that "*Overall best performance of the 1D-Var is found with bias-correction applied on channels 1-10 and a dedicated fog B. This configuration is used in the following sections.*" Taking these two statements together one can deduct that this corresponds to config 3, however it would be easier to explicitly state this at the end of section 4 (i.e. say: "configuration 3 is used in the following sections"

However later on, in Fig. 5, I see that the background configuration reappears in the right panel. *Is this necessary?* It has already been established that this configuration should not be used. In addition, by just looking at the middle and right panels of Fig. 5 the differences appear to be really minimal.

The discussion of Fig 5 is not clear. The text says at lines 240 "*We can note the large temperature warming by up to 5 K from 0 to 12 UTC during the whole fog event (in the model space) in the 0-500 m vertical range.*"

By "model space" the authors mean the left panel (i.e. forecast?). If I look at it is not clear what is meant by the large warming from 0 to 12 UTC. Is this the sharp increase in the temperature above ~100-200 m. *Is this a visual product of the rainbow color scale used? I wonder if it would be more realistic to use a continuous color scale for these plots.*

Fig 6 Is there a reason why the 1D-var overestimates specific humidity between 4 and 9 UTC? Are the brightness temperatures affected?

Fig 7 is time UTC?

Fig. 8 and related discussion. Does the introduction of the MWR data improved the statistics of undetected and false alarm? I see that the temperature errors are reduced, but is the number of false alarms and missed detections the same?

Fig. 12 x axis title is missing

Fig 13 axis labels are missing, and fonts are very small

---

## Author Comment (AC1) · 24 Jul 2020

**Manuscript review amt-2020-166 « Improvement of numerical weather prediction model analysis during fog conditions through the assimilation of ground-based microwave radiometer observations: a 1D-Var study ».**

First of all, the authors thank the two anonymous reviewers for their positive feedback and helpful comments to improve the manuscript. All modifications have been taken into account ; most relevant are highlighted in red in the new version. We hope this new version will be suitable for publication.

Reviewer 1 :

*Abstract: The authors should mention already in the abstract that the brightness temperatures are assimilated directly over a forward operator (RTTOVgb)*

→ Has been clarified in the abstract :
To that end, temperature, humidity and liquid water path (LWP) retrievals have been performed by directly assimilating brighness temperatures using a one-dimensional variational technique (1D-Var).

*Line 31-33: This formulation is not exact: it is true that satellite data provide limited information on the ABL, but not because of the complexity of data assimilation over lands, to be exact this issue makes the use of the data for NWP more difficult. Please rephrase.*

→ Thanks for pointing this inconsistency. This has been modified :
Even if satellite data provide a global coverage all over the world, they provide limited information on the ABL due to the attenuation by clouds and degraded vertical resolution in the ABL. Additionally, uncertainties in surface properties (such as skin temperature and emissivity , Guedj et al. (2011)) limit the assimilation of surface-sensitive channels over lands.

*Line 44: it could be emphasized here that the study by Otkin and Hartung et al. (2011) with 140 MWRs was an OSSE ( in contrast to your study using real data)*

→ In order to highlight this aspect we used the term « simulated network of 140 MWRs » but to make it more clear we added :
The impact of a simulated network of 140 MWRs through an OSSE was also investigated by Otkin et al. (2011) and Hartung et al. (2011) on a winter storm case.

*Line 123: For clarification for readers not familiar with (MW) remote sensing you could add half a sentence why transparent channels are omitted at low elevation angles*

→ this explanation which was given in the initial manuscript line 174 in section 3.2 has been moved to line 125 :
Transparent channels are not used at low elevation angles due to the violation of the assumption of horizontal homogeneity

*Line 134: The authors could add a sentence, why it is inadequate for fog areas*

→ this has been clarified :
As demonstrated by Ménétrier and Montmerle (2011), climatological covariances are inadequate for fog areas which exhibit a much stronger positive coupling between temperature and humidity and attenuated vertical correlations above the fog layer.

The quality of the fog B matrix (which is fixed for all fog events through the 6 month period) can depend on how many grid points in fog conditions are found within the domain (to avoid sub-sampling problem). It will also depend on how much variability in the different fog cases were taken into account. The quality of this B matrix can thus depend on the assimilation cycle which is used for its calculation. This is why we first calculated fog B matrices at different assimilation cycles. Then we run the 1D-Var algorithms and chose the fog B matrix which was giving the best RMSE with respect to radiosounding. The sentence has been rephrased to clarify.

Several fog B matrices have been computed using different assimilation cycles. The fog B matrix showing the best results in terms of root-mean-square-errors (RMSE) with respect to radiosoundings has then been selected for this study.

*Section 3.2: This section is not structured well. First it is about obs errors, then bias correction, then obs errors, then both. .*

We tried our best to make this section better structured. To that end we sub-divided this section into section 3.2.1 commenting only the results in line with the B matrix and section 3.2.2 commenting only the results in line with the bias correction. Table 4 and figure3 are used in both sections but only to comment the corresponding results.

*Line 176: what do the authors mean by the "individual errors which were added in quadrature"? Not really clear to me.*

→ it has been clarified in the manuscript :

$\sigma_{tot} = \sqrt{\sigma_{noise}^2 + \sigma_{calib}^2 + \sigma_{FM}^2}$
with $\sigma_{tot}$ the total observation errors, $\sigma_{noise}$ the uncertainty due to noise, $\sigma_{calib}$ calibration uncertainties and $\sigma_{FM}$ the uncertainty due to spectroscopic errors in the radiative transfer model.

*Line 214-215: not fully clear. . . So the authors want to say the dataset consists of stratus clouds, profiles with fog, and some clear-sky profiles?*

→ Correct, we just want to highlight that the RS profiles were launched during IOPs when we were expecting fog or stratus lowering. Thus, some of them are clear-sky, a few are under fog conditions and others in stratus-cloud. We clarified the sentence :
Radiosondes were launched during IOPs in different atmospheric conditions: the majority are under stratus-cloud and fog conditions and a few of them in clear-sky

*Line 251-256: Please give more explanations in this paragraph, why the underestima-tion of specific humidity at nighttime is due to an overestimation of saturation, ...*

→ As explained at the beginning of section 4, the AROME model predicts a thick fog layer whereas the observations (in-situ tower measurements) show fog no more than 10 m thick. Thus, we know the AROME model overestimates the saturation. The true temperature should be higher and should not reach saturation. , water would stay in its gas phase instead of being converted into liquid . What we observe is that the model converts too much water vapour into liquid (erroneously). The specific humidity is thus underestimated. One sentence has been added in the manuscript :

Indeed, as the fog layer was thicker in AROME than in the observations, we believe the model converts too much water vapour into liquid erroneously, which makes it underestimate specific humidity

*...and why most of the model increments are produced by the B matrix cross-covariances ?*

→ As concluded in section 3.2, the configuration 3 is used in the next sections. It means a block-diagonal B matrix is used under no fog conditions and a fog B matrix with cross-covariances is used when there is fog in the observation. From the visibility measurement of figure 4, we see that fog is only observed at 0 UTC and then between 5 and 9 UTC.
It means that a fog B matrix with cross-covariances is used at 0 UTC and then between 5 and 9 UTC. Outside of this time period, a diagonal B matrix is used. In figure 6, we can see that the specific humidity after 1D-Var is almost identical to the background every time a diagonal B matrix is used.
The larger increments observed during the fog events are thus attributed to the cross-covariances between temperature and humidity. We clarified this point :
This is likely due to the use of the cross correlated fog B matrix under these conditions, as opposite to the use of a block diagonal B matrix when fog is not observed.

*L264: For clarification his could be rephrased to ". . . During the period where the model fails to simulate the stratus cloud, the LWP is significantly increased in the 1D analysis with values between. . ."*

→ Agree, it has been modified

*L279: The authors could add one sentence on what the visibility diagnosis is based.*

A new sentence has been added to the manuscript to be a bit more explicative. The full explanation about this visibility diagnosis will be discussed in the manuscript of Dombrowski-Etchevers et al. (2020) :

*In this new diagnosis, the visibility is directly deduced from the liquid water content at ground. It was computed through a statistical regression between hourly maximum of liquid water content forecast by AROME and observed minimum of visibility on 100 ground stations during five months.*

*L317-318: This is not clear. Does it mean the profiles used are not forecasts but taken from an analysis with conventional data already assimilated?*

→ Currently in the Météo-France 3D-Var scheme control variables are : temperature and surface pressure, specific humidity, wind. Thus at each assimilation cycle only these variables are updated to match all the observations assimilated. However, the hydrometeors are kept unchanged.
It means that the analysis state of the AROME model for the hydrometeors correspond to the previous background, which is a 1 hour forecast. The hydrometeors are then balanced with the other fields in the first time steps of the forecast through the model physics.
This has been clarified in the manuscript :
*In fact, as hydrometeors are currently not included in the control variables of most operational variational data assimilation schemes, these fields are kept unchanged during the analysis. Thus, the analyzed hydrometeor fields correspond to the previous background. Consequently, in the following statistics, the background values of LWP correspond in fact to the LWP in the operational AROME analysis. These fields are then modified according to the updated temperature and humidity analyses in the first time steps of the forecast through the model physics.*

**Technical corrections**

*L31: Better: ". . . wich is undersampled by observations. "Even though satellite data provide a global coverage. . ."*

→ Agree and modified

*L48: Better: impact of this network was found to be neutral.."*
→ Agree and modified

*L51: Better: "AROME model with a one-dimensional.."*
→ Agree and modified

*L55: correct to: ". . . and evaluates the impact. . .."*
→ Agree and modified

*L122: correct to: ". . . consists of.."*
→ Agree and modified

*L129: replace "spatially" by "horizontally" (because spatially comprises vertical and horizontal directions)*
→ Agree and modified

*L144: Better: "...with a horizontal resolution set to 3.2km and . . ." ("finally" should be omitted)*
→ Agree and modified

*L167: no comma here*
→ Agree and modified

*L167: better: "but also on an adequate specification of. . ."*
→ Agree and modified

*Line 201: do you mean Config1 here?*
→ Correct, thanks for pointing this error

*General: References to figures in the text should be with capital "F". "Figure X" instead of "figure X".*
→ modified

*L222: Typo: "almost"*
->This sentence has been removed in the new version but thanks for pointing the error

*L229: Typo: cloud base height*
→ agree modified

*L232: better: ". . . fog is observed at 10m altitude during 40 minutes at midnight and. . ."*
→ agree modified

*L257: better: ". . . by night leads to the effect that the fog layer is not saturated any more in agreement. . ."*

→ agree modified

*L262-64: Better: ". . . with a maximum reaching 90gm-2 at 7UTC. This value, however, decreased down to. . ."*
→ agree modified

*L269: Better: "While the previous focuses on an extreme. . ."*
→ has been modified into :  While the previous **section** focuses on an extreme

*L382: Better: ". . . has been investigated with. . ."*
→ agree and modified

*L429: Better: ". . . on temperature and LWP and small but. . ."*
→ agree and modified

*Figure 8 Caption: should be re-phrased to: ". . . differences compared to tower measurements. . ."*
→ agree and modified

*Figure 13: The axes are difficult to read. Maybe the figure could be enlarged to improve this.*
→ The figure has been made again to make it more readable

---

## Author Comment (AC2) · 24 Jul 2020

**Manuscript review amt-2020-166 « Improvement of numerical weather prediction model analysis during fog conditions through the assimilation of ground-based microwave radiometer observations: a 1D-Var study ».**

First of all, the authors thank the two anonymous reviewers for their positive feedback and helpful comments to improve the manuscript. All modifications have been taken into account ; most relevant are highlighted in red in the new version. We hope this new version will be suitable for publication.

**Reviewer 2 :**

*Major comment:*
*I understand the discussion about optimal estimation in section 2.3 and 3.1, however I am a little bit perplexed by the discussion of the background covariance in section 3.2.*
*From what I could understand the a priori vector Xb = (T, Q, LWP) used in the convergence scheme specified in line 110 is provided by 1 hr AROME forecast profiles.*
*The corresponding background covariance B associated with this a priori estimate was estimated as described in line 145-150 for all cases and for a subset of fog cases and the diagonal terms were multiplied by 0.7. Where I get lost is the next section (3.2) where the background covariance is modified in a seemingly arbitrary fashion by removing the cross-correlation terms from the climatology. I entirely understand using a fog covariance for the fog cases and a climatology covariance for the non-fog case. However, to "choose" the background covariance that optimizes the retrieval results seems a little bit unorthodox.*
*My point is that the covariance should be an "objective" way (as far as possible) to quantify the uncertainty associated with the a priori information. It seems that if the background covariance associated with the Xb is not good enough for the retrieval perhaps a different choice for the a priori Xb should be made (i.e. not from the model but perhaps from a radiosonde ensemble). Alternatively, the convergence could be controlled with a multiplicative factor to B (usually called g) that is reduced at each iteration based on the behavior of the cost function. This approach is mostly used for infrared retrievals, but, in this case, it may prove beneficial as well. So perhaps I am not entirely understanding this part, in which case this procedure of "choosing" B based on the retrieval results could be better justified or may be the straight optimal estimation approach should be modified the way mentioned above.*

We would like to thank the reviewer for this very interesting question. It seems that our original paper did not justify enough the approach of possibly zeroing the cross-correlations between temperature and humidity, and this is an important point. We shall first remind that the treatment of humidity in 3Dvar has received a long attention in the NWP community (see e.g., Dee and Da Silva MWR 2003). In many operational schemes, the treatment of humidity has long been univariate (e.g. zeroing or not computing the cross-covariance between humidity and the other variables). This is for instance still the case in global ARPEGE 4DVar and in various 3DVars (eg., Barker et al MWR 2004 for the WRF model). One explanation is that the cross-covariances between humidity and temperature is flow-dependent. To be more specific, Holm et al (2002) have shown with the global IFS model that even the sign of t-q correlations was changing, being negative in average but positive in saturated conditions. Dee et al (2002) also confirmed that, if observations are not as abundant for temperature and humidity, it is more accurate to neglect mixing ratio–temperature error covariances. Whatever the used background profile (from a NWP model or from radiosounding climatology), if the analysis increment is mainly driven by the coupling, using fixed covariances for different types of weather conditions will potentially degrade the analysis. In our case, keeping cross-covariances between temperature and humidity in the climatological B-matrix is therefore suboptimal and possibly not as good as using a univariate approach for humidity,

depending on the prevalence of saturated conditions. Therefore, we consider the zeroing of temperature-humidity covariances in the climatology as a natural approach, which is of course outperformed by the use of the ensemble that brings this lacking information about saturation during fog conditions. A clean way to make the B matrix flow dependent through the 6-month study, would have been to use the AROME ensemble data assimilation (EDA) to extract hourly B matrices depending on the weather conditions for each new retrieval. Unfortunately, the AROME EDA was still in development at the beginning of the study. Only a few days were post-processed and the choice has been made to select the IOP1 with interesting fog events in the observations in order to focus on the impact of an optimal B matrix during fog conditions. As this paper mainly focuses on fog retrievals, optimizing the B matrix for all other weather conditions is beyond the scope of the paper. As an alternative approach, for non fog conditions, we decided to follow the approach still used operationally in our 4D-Var scheme by zeroing the cross-correlations between temperature and humidity to avoid the degradation in the retrievals of specific humidity.

In order to clarify this aspect ,section 3.2 has been modified introducing the explanation on why different B matrices are evaluated :

As cross-covariances highly depend on the weather conditions (Hólm et al. (2002), Michel et al. (2011)) and the use of fixed covariances is not optimal when dealing with different atmospheric scenario, Config1 aims at evaluating the impact of the cross-correlations between temperature and humidity on the retrievals. To that end Config1 corresponds to the same configuration but removing the cross-correlations between temperature and specific humidity. It can be noted that this approach is still used in various 3D/4D-Var operational schemes (Barker et al. (2004)).

We also included an additional paragraph at the end of the section to clarify the conclusion :

Figure 3 confirms that the best configuration in terms of B matrix corresponds to Config2 compared to the CTRL configuration. In fact, the use of a "climatological" B matrix with cross-correlations degrades both temperature and humidity retrievals but more significantly specific humidity up to 4 km. Overall, these results confirm that, for MWRs, humidity increments in the lowest levels are significantly driven by the cross-correlations between temperature and humidity. These correlations (sign and amplitude) being highly dependent on the weather conditions, the B matrix should ideally be updated for each profile. When it is not possible, the use of a block diagonal B matrix might be preferable to avoid degradation in the retrievals due to inaccurate cross-correlations. This result is in line with the study of Dee et al 2002 which showed that, when humidity is less adequately observed than temperature, it is more accurate to neglect humidity – temperature error covariances. However, when an adapted flow-dependent B matrix is used, the specific humidity analysis is improved. In the future, the use of ensemble data assimilation schemes should enable deriving optimal B matrices evolving in time and space to be consistent with the weather conditions.

*Minor comments:*
*Abstract: There is terminology that is not defined for example, in lines 11, 12, 15, what are 1D-var increments? I suggest either making the abstract less detailed about the results or defining the terms used.*

→ We have removed some details from the abstract and we do not use the word « increment » anymore. We defined this term later in section 2.3 :
Through the manuscript, the atmospheric state minimizing the cost function is called the "analysis" ($x_a$ ) and "increment" refers to the difference between the a priori $x_b$  and the analysis.

*Table 4 is not clear. The caption says "Error reduction (%)" over the background. It is not clear what the negative number means. Does it mean that the retrieval is actually increasing the RMSE with respect to the background?*

RMSE are always computed with respect to in-situ measurements on the tower.
Then two RMSE are computed :
RMSE_xb : errors of the background with respect to the tower
RMSE_xa : errors of the analysis with respect to the tower
What is called error reduction in the manuscript is defined by :
ER=1-RMSExa/RMSExb

when this number is negative it means that the retrievals increase the errors with respect to the tower measurements compared to the background (the 1D-Var retrieval is thus worse than the background statistically).

We modified the table 4 caption to clarify :
Reduction in the RMSE with respect to tower measurements after the 1D-Var analysis (RMSExa ) compared to the background (RMSExb) for all weather conditions (upper part) or only fog events (lower part) : ER = 1− RMSExa/RMSExb (%)

*Table 4: Just to make sure I understand correctly, this statistic is computed by taking one layer of the retrieval profile grid corresponding to the tower height of 50 and one layer corresponding to 120 m? I would imagine that the retrievals at these two layers are highly correlated because the vertical resolution of the radiometer at this height is about 100 m. Therefore, is there even a merit to look at two layers vs. averaging the tower and radiometer measurements between 50 and 120m?*

→ We agree with the reviewer that the two retrievals are potentially highly correlated. This is now stated in the following sentence, reported at line 179 of the revised manuscript. However, we prefer to keep the comparison at both levels, as it conveys some information, e.g., that the reduction in T(K) is higher at the lower level.

It is important to note that given the relative low vertical resolution of MWR retrievals, the retrievals at 50 and 120 m are likely to be highly correlated.

*Fig. 3 It is not clear what "closest GP B clim" mean in the labels.*

→ We agree and have made labels consistent with the names of the different configurations defined in table1.

*However later on, in Fig. 5, I see that the background configuration reappears in the right panel. Is this necessary? It has already been established that this configuration should not be used. In addition, by just looking at the middle and right panels of Fig. 5 the differences appear to be really minimal.*

→ As the largest differences are limited in time (only when a correlated fog B matrix is used for humidity between 4 and 9 UTC) and vertical spread (the first 500m), it is true that the differences can appear minimal when looking globally at the figure. To avoid confusion, we decided to remove the control configuration in the right panels and show only the optimal configuration 3.

*The discussion of Fig 5 is not clear. The text says at lines 240 "We can note the large temperature warming by up to 5 K from 0 to 12 UTC during the whole fog event (in the model space) in the 0-500 m vertical range." By "model space" the authors mean the left panel (i.e. forecast?). If I look at it is not clear what is meant by the large warming from 0 to 12 UTC. Is this the sharp increase in the temperature*

*above ~100-200 m. Is this a visual product of the rainbow color scale used? I wonder if it would be more realistic to use a continuous color scale for these plots.*

→ « In the model space » means when the model simulates fog. It only refers to the time period 0 to 12 UTC which corresponds to fog only in the model simulation. In the observation fog is only observed between 4 and 9 UTC. If we look at the temperature between 0 and 12 UTC in the first ~ 200 meters, we can see that the temperature is much colder on the left panel (the model background) compared to the middle planel (1D-Var). We just wanted to stress that this time window corresponds to the simulated fog event (and not the time where the fog was actually observed). In order to make the figure more readable, we limited the y axis to the 0-1 km range and we clarified the text :

We can note the large temperature increment, up to 5 K from 0 to 12 UTC essentially in the first 250 m, after 1D-Var is applied ; this is the period where the model simulates a thick fog event not confirmed by the observations.

*Fig 6 Is there a reason why the 1D-var overestimates specific humidity between 4 and 9 UTC? Are the brightness temperatures affected?*

→ The most reasonable explanation is that the positive cross-correlations between temperature and humidity are probably over-estimated. However, we are probably within the retrieval uncertainty. One sentence has been added to the manuscript :

Though closer to the in-situ observations, 1D-Var retrievals slightly overestimate specific humidity between 4 and 9 UTC. This is most likely due to over-estimated positive cross-correlations between temperature and humidity in the B matrix

*Fig 7 is time UTC?*
→ Correct, this has been modified in the figure

*Fig. 8 and related discussion. Does the introduction of the MWR data improved the statistics of undetected and false alarm? I see that the temperature errors are reduced, but is the number of false alarms and missed detections the same?*

→ This is a very good question that we investigated. However, as the 1D-Var only deals with the integrated liquid water (LWP) and does not have enough information with the MWR alone to correctly localize several cloud layers, it is complicated to interpret new statistics based only on the 1D-Var analysis. In fact, if there is already a fog or a cloud layer, the 1D-Var will normalize the vertical distribution so that the integrated path is closer to the observation. If the model initially gives no fog or cloud whereas the observation is cloudy, the 1D-Var will add a new fog layer at the level of maximum relative humidity (which may or may not be close to the ground). The scores to determine false alarms, missed events and good detections are, on the other hand, based only on the LWC at the ground.
It means that if there is a cloud layer on top of the fog, the LWP might be increased erroneously at the ground due to the cloud aloft during false alarms. If we look at the background profiles during false alarms, only 33 % do not have cloud detected by the ceilometer. It means that in the remaining 67 %, the model simulates fog at the ground though it is not detected by the visibility sensor and a cloud aloft is observed in the measurements. For these cases, the 1D-Var cannot remove the liquid water content at ground (as there is a cloud aloft) and might even increase the LWC. If we just evaluate the 1D-Var retrievals based on the false alarms scores we will conclude that there is either no impact or even a degradation. The only way to really make a conclusion would be to let the

model physics get the liquid water content profile balanced with the new temperature and humidity profiles before recomputing the score.

The same problem occurs when evaluating missed fog events if there is not liquid water in the background at the ground. However, for this specific study, most of the background profiles have a small amount of liquid water at ground level, even though the visibility at ground is not reduced to less than 1000 m. The rate of missed fog events is decreased from 27 % in the background to 19 % in the 1D-Var analysis. Again this evaluation is only based on the LWC change at the ground and it would be interesting to evaluate the impact of the new temperature and humidity fields on the LWC after a few time steps of forecasts but this is beyond the scope of this paper. This investigation into the forecast impact will be studied within the framework of the SOFOG3D experiment.

We included a discussion on this topic in the new version of the manuscript at the end of section 5 :

The next natural step of this study would be to calculate updated scores of fog detections with the new 1D-Var analyses compared to the background profiles. However, forecast scores are only based on the LWC at ground whereas the 1D-Var works on the liquid water path without information on the cloud vertical structure. During false alarms, conclusions on the impact on forecast scores are complexified by the presence of cloud layers above fog in a majority of false alarms which can cause an increase in LWC at ground. As for the hit ratio,  it is increased from 73 % in the background to 81 % in the analysis. The rate of missed fog events is also decreased from 27 % in the background to 19 % in the 1D-Var analysis. However, as this evaluation is only based on the LWC change at the ground, it is necessary to evaluate the impact of the new temperature and humidity fields on the LWC after a few time steps of forecasts but this is beyond the scope of this paper. This investigation into the forecast impact will be studied in the future within the framework of the SOFOG3D experiment (section 6).

*Fig. 12 x axis title is missing :*
→ Lables for x axis are now reported  in addition to the figure caption.

*Fig 13 axis labels are missing, and fonts are very small*
→ We have recomputed the figure with labels and titles increased.

---

## Author Comment (AC4) · 24 Jul 2020

The comment was uploaded in the form of a supplement:
https://amt.copernicus.org/preprints/amt-2020-166/amt-2020-166-AC4-
supplement.pdf
* * *

---

## Author Response (AR2)

**Manuscript review amt-2020-166 « Improvement of numerical weather prediction model analysis during fog conditions through the assimilation of ground-based microwave radiometer observations: a 1D-Var study ».**

First of all, the authors thank the two anonymous reviewers and the editor for their positive feedback on this new version which allows the publication with minor reviews.
All the suggested modifications have been taken into account and are highlighted in red in the marked-up manuscrit.

[revised manuscript text omitted]